# Affinity and dose of TCR engagement yield proportional enhancer and gene activity in CD4+ T cells

Karmel A Allison[1,2], Eniko Sajti[3,4], Jana G Collier[1], David Gosselin[1], Ty Dale Troutman[1], Erica L Stone[5,6], Stephen M Hedrick[1,5]*, Christopher K Glass[1,7]*

[1]Department of Cellular and Molecular Medicine, University of California, San Diego, United States; [2]Bioinformatics and Systems Biology Program, University of California, San Diego, United States; [3]Department of Pediatrics, University of California, San Diego, United States; [4]Rady Children's Hospital, San Diego, United States; [5]Molecular Biology Section, Division of Biological Science, University of California, San Diego, United States; [6]Translational Tumor Immunology Program, Wistar Institute Cancer Center, Philadelphia, United States; [7]Department of Medicine, University of California, San Diego, United States

**Abstract** Affinity and dose of T cell receptor (TCR) interaction with antigens govern the magnitude of CD4+ T cell responses, but questions remain regarding the quantitative translation of TCR engagement into downstream signals. We find that while the response of mouse CD4+ T cells to antigenic stimulation is bimodal, activated cells exhibit analog responses proportional to signal strength. Gene expression output reflects TCR signal strength, providing a signature of T cell activation. Expression changes rely on a pre-established enhancer landscape and quantitative acetylation at AP-1 binding sites. Finally, we show that graded expression of activation genes depends on ERK pathway activation, suggesting that an ERK-AP-1 axis plays an important role in translating TCR signal strength into proportional activation of enhancers and genes essential for T cell function.

*For correspondence: shedrick@ucsd.edu (SMH); cglass@ucsd.edu (CKG)

## Introduction

The question of how the T cell receptors (TCRs) of CD4+ T cells respond to ligands of differing affinities and concentrations with such remarkable specificity is of great interest to the study of immunity. The TCR binds to antigen presented by the molecules encoded by the Major Histocompatibility Complex (MHC) such that strength of the TCR signal in response to a particular peptide-MHC complex (pMHC) is dependent on all three components– the antigenic peptide, the MHC, and the TCR itself (*Heber-Katz et al., 1982*). Variations in signal efficiency are thus caused by the generated TCR sequence (*Hedrick et al., 1984*; *Jerne, 1971*), genetic differences in MHC (*Heber-Katz et al., 1982*; *Hedrick et al., 1982*), and the peptide being presented (*Solinger et al., 1979*). Even small differences in the number or affinity of these pMHC-TCR interactions are read by the TCR and have important consequences for the nature and extent CD4+ T cell activation; high-affinity interactions lead to inflammatory responses at a lower concentration of antigen, increased Interleukin 2 (IL2) and IFNγ production, and increased proliferation (*Heber-Katz et al., 1982*; *Hedrick et al., 1982*; *Solinger et al., 1979*; *Alexander, 1993*; *Rogers and Peptide dose, 1999*; *Rogers et al., 1998*; *Sloan-Lancaster et al., 1993*; *Tubo et al., 2013*), whereas lower affinity interactions can lead to incomplete phosphorylation of downstream signaling complexes (*Kersh et al., 1998b*; *Sloan-*

**eLife digest** T helper cells recognize and respond to bacteria, viruses and other invading microbes and thus play a central role in the adaptive immune system. These cells have a receptor on their surface that binds to fragments of proteins – known as oligopeptides – from the microbes that have been digested and presented on the surfaces of other immune cells. Once active, T helper cells multiply, grow and release signals that regulate genes in other cells to promote immune responses. Previous studies suggest that a T helper cell's response is binary – that is, either on or off. However, this does not explain how the strength of the T cell response to infection can vary.

Allison et al. used a technique called high-throughput sequencing to examine the activity of genes in T helper cells from mice that had been genetically engineered to only produce one type of T cell receptor. For the experiments, the T cells were exposed to various concentrations of different peptides known to bind either well or poorly to the receptor. Allison et al. found that, once activated, the response of an individual T cell was not binary, but instead was related to the strength of the signal it received through its receptor.

Further experiments showed that although a subset of the genes activated in T helper cells do respond in a binary fashion, the activities of many other genes involved in immune responses and cell metabolism were related to the strength of the signal from the receptor. This "analog" gene activation depends on the level of activity of the MAP kinase signaling pathway. Together, Allison et al.'s findings help us to understand how T cells are able to fine-tune immune responses to invading microbes. The next challenge will be to investigate the mechanisms underlying binary and analog gene activity in T cells.

*Lancaster et al., 1994*), anergy (*Sloan-Lancaster et al., 1993*, *1994*), or TCR antagonism (*Alexander, 1993*; *Kersh et al., 1998b*). The precise result of low-affinity engagement varies with experimental conditions, but in each case, a cellular phenotype distinct from high-affinity engagement is produced.

There exists a well-characterized model system for studying the effects of ligand affinity on CD4+ T cell activation: the AND mouse is a strain with a transgenic CD4+ T cell TCR (*Kaye et al., 1989*). This TCR recognizes pigeon cytochrome *c* (PCC) along with synthetic and species-variant cytochrome *c* oligopeptides (*Solinger et al., 1979*; *Rogers and Peptide dose, 1999*; *Rogers et al., 1998*). Notably, though many of the peptides differ from PCC by a single amino acid, the effects of TCR recognition of the peptides vary greatly. Kinetic parameters and cytokine output of the interaction with many cytochrome *c* peptides and their analogues have been described (*Rogers and Peptide dose, 1999*; *Rogers et al., 1998*). Differences in microcluster formation at the membrane have likewise been described (*Varma et al., 2013*).

These variable responses to ligands of differing affinity are especially interesting in the context of the digital TCR response. TCR responses have been characterized as digital (*Coward et al., 2010*)—that is, signaling downstream of the TCR is either all-on or all-off, such that a given T cell must either be committed to a full response or to no response. Previous work has established this switch-like behavior as observable in terms of extracellular markers such as CD69 (*Das et al., 2009*; *Daniels et al., 2006*), ERK pathway component localization (*Das et al., 2009*; *Daniels et al., 2006*; *Prasad et al., 2009*), NF-κB activation (*Kingeter et al., 2010*), NFAT localization (*Marangoni et al., 2013*; *Podtschaske et al., 2007*), cell-cycle entry (*Au-Yeung et al., 2014*), and cytokine production (*Podtschaske et al., 2007*; *Huang et al., 2013*). As a result, differences in the magnitude of responses to ligands of varying affinity would be attributed to greater frequencies of T cells responding at the population level, rather than per-cell variability (*Au-Yeung et al., 2014*; *Huang et al., 2013*; *Zikherman and Au-Yeung, 2015*; *Butler et al., 2013*). Still, some aspects of the TCR response have been described as analog, or varying in proportion to the strength of signaling: CD3ζ chain phosphorylation (*Kersh et al., 1998a*; *Sloan-Lancaster et al., 1994*; *Daniels et al., 2006*; *Kersh et al., 1999*; *Kersh et al., 1998a*); Zap70 activation (*Daniels et al., 2006*; *Prasad et al., 2009*); intracellular calcium concentrations (*Irvine et al., 2002*); expression of the transcription factor

IRF4 (*Man et al., 2013*; *Nayar, 2014*); and cell division time (*Marchingo, 2014*). It is unclear how these analog components of the TCR response fit in to a digital model.

Both the ability of the TCR to discriminate with high resolution between ligands and the digital nature of TCR signaling have been extensively studied at the level of signaling. Downstream of the TCR, a number of signaling pathways govern the molecular response to engagement, allowing T cells to grow, divide, and acquire immune effector functions consistent with the inciting stimulus (*Murphy and Blenis, 2006*; *O'Sullivan and Immunology, 2015*; *Proud, 2007*; *Santamaria and Ortega, 2006*; *Wang and Green, 2012*). AKT and PKCθ interact at the cell membrane and jointly serve to induce nuclear translocation of the pro-inflammatory transcription factor NF-κB, which in turn is able to activate target genes (*Huang and Wange, 2004*). In particular, AP-1, which comprises homo- or heterodimers assembled from proteins of the Fos, Jun, and ATF transcription factor families (*Murphy et al., 2013*), requires both TCR and co-stimulatory signaling (*Rincón and Flavell, 1994*), and it is usually activated by the Ras/Raf/Mek/Erk pathway (*Murphy and Blenis, 2006*; *Schade and Cutting edge, 2004*). At least four feedback loops have been identified in thymocytes and peripheral T cells downstream of the TCR (*Coward et al., 2010*; *Feinerman et al., 2008*). Collectively, these feedback loops serve to enforce a digital response by either dampening sub-threshold signaling or amplifying above-threshold signaling, resulting in T cell responses that are all-off or all-on, respectively.

Despite these insights into the signaling pathways downstream of TCR activation, there is little known about the transcriptional programs that determine the distinct phenotypes resulting from high- versus low-affinity stimulation. In this study, we address the question of affinity at the level of the chromatin. We take advantage of the PCC system to assess the effects of varying the dose and affinity of peptide presentation to CD4+ T cells on enhancer formation and gene expression, giving us a genome-wide picture of how TCR signaling is able to achieve such highly specific responses despite its apparent digital signaling pattern. We find first that the digital/analog dichotomy is too simple, and instead CD4+ T cells respond to ligands of varying dose and affinity by modulating both the frequency of responding cells and the level of activation of responding cells at a single cell level. In other words, activation markers are analog with respect to the strength of TCR signaling when comparing across doses and affinities, but for any single dose and affinity, the overall signaling pattern is digital for the population of cells. We next show that at the population level, the combined effects of analog precision and increasing frequency of responder cells produce gene expression patterns that directly reflect the strength of TCR signaling for a set of activation signature genes. These gene expression patterns can be used to assess CD4+ T cell activation status, and we develop a tool for ranking arbitrary CD4+ T cell populations by activation score. Underlying these gene expression patterns, we find that the enhancer landscape is largely pre-existing, such that TCR engagement results in activation of primed enhancers rather than through selection of new enhancers. Finally, we show that the graded activation score and the expression of activation signature genes are dependent on the amount of phosphorylated ERK activity downstream of the TCR. Together, these results suggest that the degree of ERK activation translates the analog TCR signal resulting from varying the dose and affinity of TCR engagement into downstream gene expression programs.

## Results

### The TCR response is analog: quantitative responses to signal strength

In order to understand the effects of the digital TCR response on the transcriptional landscapes of CD4+ T cells, we first sought to characterize the 'on-state' of the TCR response. CD4+ T cells and CD11c+ antigen presenting cells (APCs) were isolated from AND transgenic mice (*Figure 1—figure supplement 1A,B*) and co-cultured for 24 hr with one of a panel of previously described (*Rogers and Peptide dose, 1999*; *Rogers et al., 1998*) peptides at several different doses. Cell activation was then measured at a single cell level using flow cytometry. As expected, for each given peptide and dose, CD4+ T cells followed a digital pattern, appearing either all-on or all-off according to extracellular activation markers such as CD69 (*Figure 1A*) and CD25 (*Figure 1B*). Increasing the peptide dose or affinity significantly increased the percent of activated cells in the population (*Figure 1C,D*). However, when we compared across peptides and concentrations, it was clear that

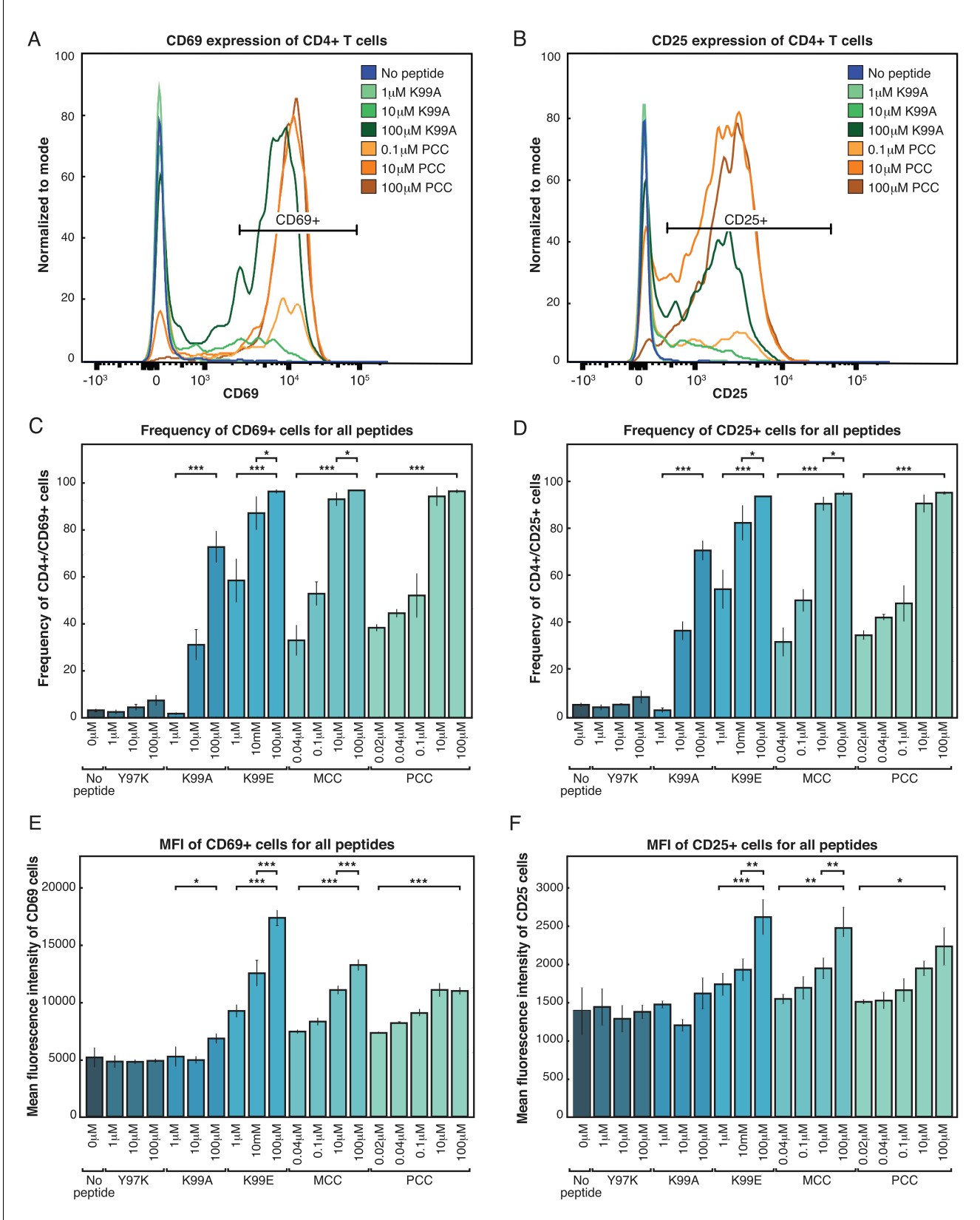

**Figure 1.** Both frequency of responding cells and per-cell activation levels increase with increasing signal strength. (**A**) Purified AND T cells and CD11c + APCs were co-cultured in the presence of indicated peptides at the indicated concentrations for 24 hr. Flow cytometry was then used to phenotype

*Figure 1 continued on next page*

*Figure 1 continued*

the CD4+ T cells. The histograms show CD69 expression of CD4+ T cells. The annotated bar indicates the gate used to identify CD69+ CD4 cells in subsequent figures. (B) Gating on CD4+ cells as in 1A, there is a bimodal distribution of CD25 expression resulting from activating levels of high-affinity (PCC) or low-affinity (K99A) peptides. The annotated bar indicates the gate used to identify CD25+ cells. (C) The percent of CD4+ cells that are CD69+ (using gate shown in 1A) varies with the peptide presented and concentration of the indicated peptide. (D) The percent of CD4+ cells that are positive for the activation marker CD25 varies with both peptide and dose. (E) Gating on CD4+ CD69+ cells (as shown in 1A), the geometric mean fluorescence intensity (MFI) of CD69 per cell in each condition varies. (F) The geometric MFI of CD25, gated on CD4+ CD25+ cells (as shown in 1C), varies with peptide and dose. (P-values based on Student's t test; *p<0.05, **p<0.01, ***p<0.001.)

The following figure supplement is available for figure 1:

**Figure supplement 1.** CD4+ T cells and APCs were purified from AND mice.

the activation level of the on-state cells was not 'all on.' Gating on CD69+ cells, each different peptide and different dose of a peptide achieved a different amount of CD69 per cell (*Figure 1E*). Gating on CD25+ cells yields similar results, with varying amounts of CD25 expressed per cell dependent on both the dose and the affinity of the stimulation (*Figure 1F*). Thus, while under a given condition the CD4+ T cells were either on or off as per classical digital signaling, when comparing across a panel of conditions, the activation level of the on-state cells is analog with respect to the strength of the TCR signal.

## Gene expression is graded genome-wide

In order to understand the effect of this variability genome-wide, we selected two peptides—the low-affinity K99A and the high-affinity PCC—and sequenced mRNA from CD4+ T cells exposed to both a low and a high concentration of each peptide. We compared the gene expression profiles at 24 hr across five conditions (no-peptide; low-dose, low-affinity (10 µM K99A); high-dose, low-affinity (100 µM K99A); low-dose, high-affinity (0.1 µM PCC); and high-dose, high-affinity (10 µM PCC)), four out of five of which displayed some degree of activation as measured by extracellular markers such as CD69 or CD25. We used principal component analysis (PCA) to determine the primary axes of variation across the approximately 3,000 genes that were expressed above ten reads per kilobase per million (RPKM) and at least two-fold different between any two conditions. A single principal component explained more than 99% of the variance in gene expression changes (*Figure 2—figure supplement 1A*).

The first principal component ranks the samples according to what would be expected based on TCR signal strength and extracellular markers such as CD69 and CD25 (*Figure 2A*). As PC1 captured the gene expression changes concomitant with increasing activation, we extracted the most positive 10% and most negative 10% of the genes along PC1 to determine which genes were important for the axis. The 10% of genes contributing the most positive signal to PC1 were increasing in a generally graded manner with TCR signal strength across the samples, and the 10% of genes contributing the most negative signal were decreasing (*Figure 2B*; two-tailed p-values based on permutation testing of 2.9e-11 and 1.8e-28, respectively).

Collectively, the most indicative 10% of genes for PC1 provide a multidimensional signal for ranking the samples in one dimension according to TCR signal strength and activation state of the CD4+ T cell; we therefore call these genes "activation signature genes." Looking more closely at the genes in this group, we see many well-documented immune response genes such as *Tbx21* (Tbet), *Stat1*, and *Tnf* (*Figure 2C*), all of which increase in a graded manner along with TCR signal strength at the population level. *Irf4*, previously reported to increase in expression in an analog manner downstream of the TCR on a per-cell basis (*Man et al., 2013*; *Nayar, 2014*), was also among the activation signature genes, and showed the same graded response pattern at the population level across the conditions (*Figure 2D*). Gene Ontology analysis yielded several enriched gene categories among the activation signature, including Myc targets, post transcriptional regulation of gene expression, MTORC1 signaling and response to cytokine and regulation of translational initiation (*Figure 2E*) exemplified by Eif3a (*Figure 2—figure supplement 1B*). Protein biosynthesis has been previously shown to be increased upon T cell activation, and here we see that many of the genes involved in increasing translational activity are themselves up-regulated in a manner that is proportional to the

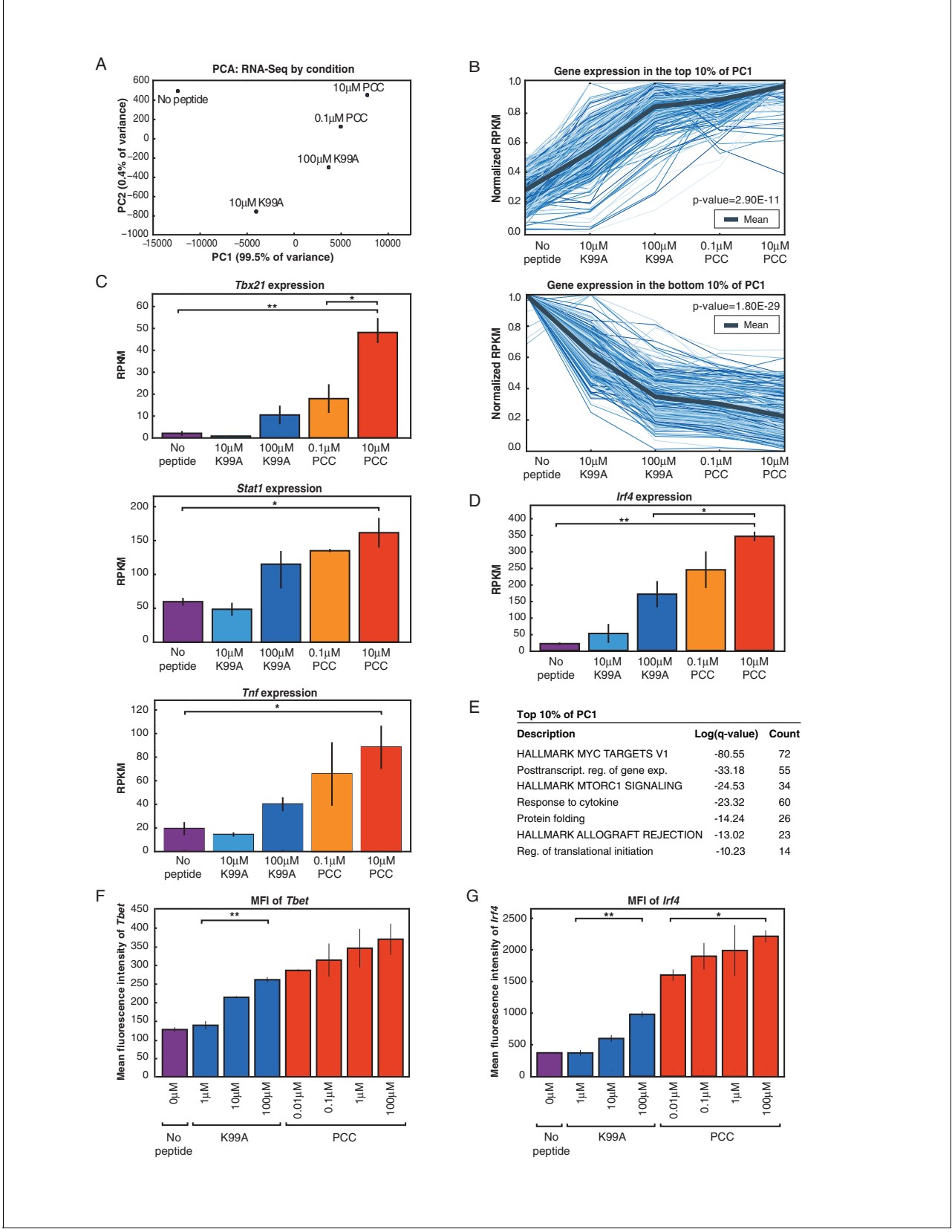

**Figure 2.** RNA-Sequencing reveals graded expression of activation signature genes. (**A**) Principal component analysis (PCA) of the approximately 3,200 genes that changed between any two samples reveals that the primary axis of variation (PC1, shown along the x-axis) orders the five conditions by

*Figure 2 continued*

increasing TCR signal strength: No Peptide; low-dose, low-affinity (10 μM K99A); high-dose, low-affinity (100 μM K99A); low-dose, high-affinity (0.1 μM PCC); and high-dose, high-affinity (10 μM PCC). (B) After ordering the ~3200 genes used for PCA by their contribution to PC1, we extracted the top 10%—that is, the ~320 genes contributing most positively to a sample's PC1 value—and the bottom 10%—that is, the ~320 genes contributing most negatively to a sample's PC1 value. Each group displays a clear trend, with the top 10% increasing in expression as signal strength increases, and the bottom 10% decreasing in expression. Each blue line represents a gene, with reads per kilobase per million (RPKM) normalized from 0 to 1 across the five conditions. Significance was determined using permutation testing, where the mean difference between genes in the No Peptide sample as compared to 10 μM PCC was normally distributed over randomly generated groups of genes. This normal distribution was compared to the top 10% and bottom 10% genes to generate a p-value. (C) Genes in the top 10% of PC1, termed activation signature genes, include many genes previously identified as important to CD4+ T cell activation, such as *Tbx21* (Tbet), *Stat1*, and *Tnf*. Reads per kilobase per million (RPKM) for each increases with increasing signaling strength. (D) *Irf4*, a transcription factor previously shown to be more highly expressed with increasing TCR affinity, shows graded expression across the five conditions. (E) Gene Ontology (GO) analysis of activation signature genes shows enrichment for protein biosynthesis and molecular chaperone genes. P-values shown are Benjamini-Hochberg adjusted p-values. (F) As measured by flow cytometry, the geometric MFI of Tbet in CD4+ cells increases on a per-cell basis with increasing signal strength. Note that MFI of the entire population of CD4+ cells is shown, as *Tbet* distribution is unimodal. (G) Similarly, per-cell protein levels of IRF4 increase with increasing signal strength when measured with flow cytometry. (p-values based on Student's t test; *p<0.05, **p<0.01, ***p<0.001.)
The following figure supplement is available for figure 2:

**Figure supplement 1.** RNA-Sequencing reveals graded expression of activation signature genes.

level of activation across the population (*Beretta, 2004*; *Bjur et al., 2013*). Another enriched ontological category was molecular chaperone genes that are responsible for protein folding and unfolding, including six of the Cct family of chaperones (for example, *Cct2*: *Figure 2—figure supplement 1C*) and five heat shock family members (for example, *Hsph1*) that increased with TCR signal strength.

The graded increase of activation signature genes at the population level corresponded with single-cell increases in CD69 and CD25 protein, but it was unclear whether the graded response of proteins associated with activation signature genes was generalizable. In order to determine whether proteins derived from activation signature genes increased on a per-cell basis in more cases, we determined the per-cell protein levels of a panel of genes from the activation signature set with flow cytometry. Not all increases in mRNA levels were reflected at the level of protein (*Figure 2—figure supplement 1D*), but for those that were, the increases in mRNA resulted in increases both in the frequency of cells responding (data not shown) and in protein levels on a per-cell basis. This is exemplified by the expression of Tbet (*Figure 2F*), IRF4 (*Figure 2G*), CD200 (*Figure 2—figure supplement 1E*), Ly6a (*Figure 2—figure supplement 1F*), and Tnsf11 (RANKL; *Figure 2—figure supplement 1G*).

## Per-cell transcription levels are analog for a subset of genes

To investigate whether graded levels of expression of activation signature genes seen at the RNA level was a reflection of per-cell activation levels or just the frequency of responder cells we conducted both RNA flow cytometry and RNA-seq for cells sorted by extracellular CD69 expression. RNA flow cytometry involves the hybridization of fluorescent probes to RNA targets within a cell, followed by analysis of per-cell fluorescence via flow cytometry. Thus, observed fluorescence intensity corresponds directly to single-cell expression levels of mRNAs of interest. Using this technique, we observed that mRNA expression levels of *Irf4* increased across the five conditions on a per-cell basis in the same manner as seen at the population level and at the protein level (*Figure 3—figure supplement 1A*). For each population, the mean fluorescence intensities of *Irf4* mRNA increased in the same manner as seen with population-level mRNA expression and single-cell protein levels (*Figure 3A*, *Figure 3—figure supplement 1B*). In contrast, beta actin mRNA as measured by this technique was constant under each treatment condition (*Figure 3—figure supplement 1C,D*). Thus, for some mRNAs, in addition to modulating the frequency of responding cells, peptide dose and affinity tune the strength of response to TCR signaling on a per-cell basis.

In order to look genome-wide, we conducted RNA-sequencing on CD4+ T cells treated as before, but sorted by the activation marker CD69, such that we selectively analyzed responder cells. In this way, we were able to control for the effect of frequency—the observed RNA-seq levels for

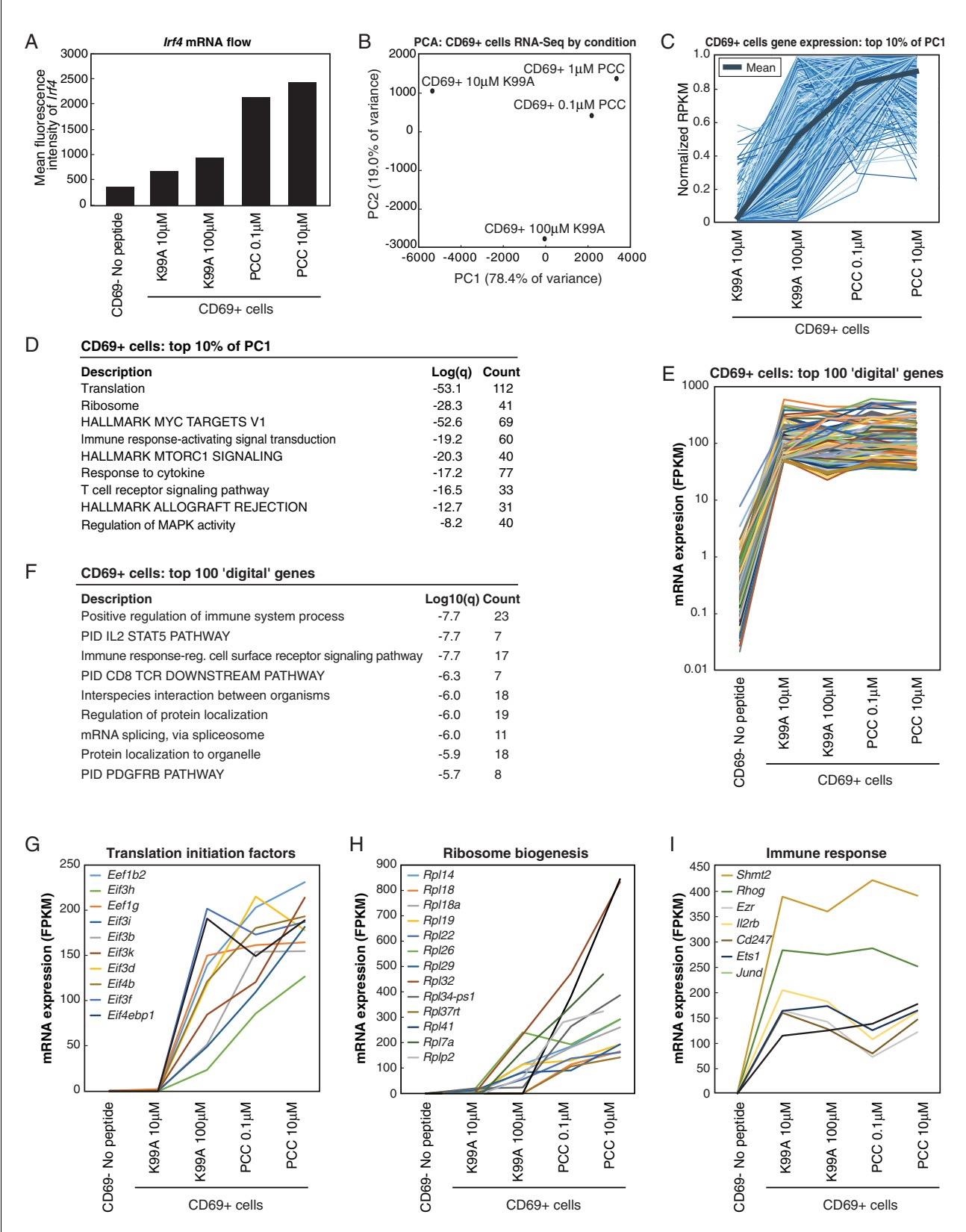

**Figure 3.** Single-cell RNA levels reflect whole-population levels for a subset of genes. (**A**) RNA flow cytometry was used to determine the single-cell levels of RNA transcripts for *Irf4* across the five peptide conditions. The mean fluorescence intensity of the *Irf4* RNA probe increases across the five

*Figure 3 continued on next page*

Figure 3 continued

conditions assayed via RNA flow cytometry. Data is representative of two technical and two biological replicates. (B) Treated cells were sorted, and RNA-sequencing was performed on the CD69+ cells from each peptide condition such that the effect of responder frequency was controlled. PCA of the genes that changed between any two samples reveals that the primary axis of variation (PC1, shown along the x-axis) orders the four conditions that had any retrievable CD69+ cells by increasing TCR signal strength: low-dose, low-affinity (10 μM K99A); high-dose, low-affinity (100 μM K99A); low-dose, high-affinity (0.1 μM PCC); and high-dose, high-affinity (10 μM PCC). (C) As with the whole-population RNA-sequencing data, looking at the top 10% of genes along PC1 revealed an expression profile that reflects the analog signal seen with external activation markers such as CD25 and CD69. Each blue line represents a gene, with reads per kilobase per million (RPKM) normalized from 0 to 1 across the five conditions. (D) Gene Ontology (GO) analysis of the genes in the top 10% of PC1 shows enrichment for terms related to metabolic processes such as translation and RNA biosynthesis. (E) There was additionally a digital cluster of genes identified that showed thressholding behavior upon cell activation. For the digital cluster, expression levels were low in the control CD69- cells, and universally higher across all CD69+ samples. The peptide conditions are sorted along the x-axis, and the normalized RPKM for the 100 most highly induced/most consistently expressed genes in the cluster is shown along the y-axis. (F) GO analysis of the genes in the digital cluster shows an enrichment for terms related to immune signaling pathways. (G) RPKM levels for several genes classified as 'translation initiation factors' are shown. For most of the genes, expression is low for the CD69- control, and increases in an analog fashion across the CD69+ cells from each treated condition. The peptide conditions are sorted along the x-axis, and RPKM for each gene is shown along the y-axis. (H) RPKM levels for several genes in the 'ribosome biogenesis' GO category are shown. Expression is low for the CD69- control, and increases in an analog fashion across the CD69+ cells from each treated condition. The peptide conditions are sorted along the x-axis, and RPKM for each gene is shown along the y-axis. (I) RPKM levels for several genes indicative of the digital cluster are shown. Expression is low for the CD69- control, but relatively level across all CD69+ samples. The peptide conditions are sorted along the x-axis, and RPKM for each gene is shown along the y-axis.

The following figure supplement is available for figure 3:

**Figure supplement 1.** RNA flow cytometry for *Irf4* shows graded increases.

each condition reflected only the activated cells, and thus were not diluted by non-responder cells. As with the whole-population RNA-seq, we used PCA to determine the primary axes of variation across the four samples for which CD69+ cells could be collected. (Insufficient responder cells could be retrieved from the no-peptide condition for sequencing.) Across the four treated samples, the first principal component again sorted the conditions according to the relative strength of TCR signaling (*Figure 3B*). Looking at the genes composing the top 10% of PC1, we saw, as with the whole-population RNA-seq, that gene expression patterns largely reflected the same trend, increasing with increasing TCR signal (*Figure 3C*, *Figure 3—figure supplement 1E*).

To assess what genes made up the activation signature seen in PC1, we performed GO analysis on the genes in the top 10% of PC1. As with the whole-population data, genes associated with cell metabolic processes like translation and RNA biosynthesis were enriched (*Figure 3D*), indicating that the expression of genes in key cellular activation pathways increases with increasing TCR signaling on a per-cell basis as well as at the population level. In contrast to the whole-population data, there was a group of genes that were near maximally activated in CD69+ cells treated with the low concentration of K99A and exhibited similar expression across all treatment conditions. This 'digital' pattern is illustrated in *Figure 3E* for the 100 most highly induced/most consistently expressed genes and for representative genes in *Figure 3—figure supplement 1F*. GO analysis demonstrated that this digital cluster of genes was enriched for immune signaling terms (*Figure 3F*). Expression profiles for representative mRNAs in the translation initiation factor and the ribosome biogenesis groups are illustrated in *Figure 3G,H* and *Figure 3—figure supplement 1E*, and representative mRNAs in the digital group are illustrated in *Figure 3I* and *Figure 3—figure supplement 1F*.

Thus, the graded increase in expression of activation signature genes at the population level is a reflection of both frequency of responding cells and incremental increases in expression levels on a single-cell level for at least a subset of genes, including key cellular metabolism genes.

## An activation score ranks CD4+ T cell samples by activation status

Given that PC1 was able to distinguish between the five conditions according to activation state, we extracted the genes from the top and bottom of the whole-population PC1 that were consistent across replicates to use as a general-purpose activation score able to correctly rank the five conditions by TCR signal (*Figure 4A*). We compared samples from several publicly available datasets, and the activation score was able to quantitatively rank conditions within a given experiment set such that activated and naïve CD4+ T cells could be distinguished and further that the effects of various

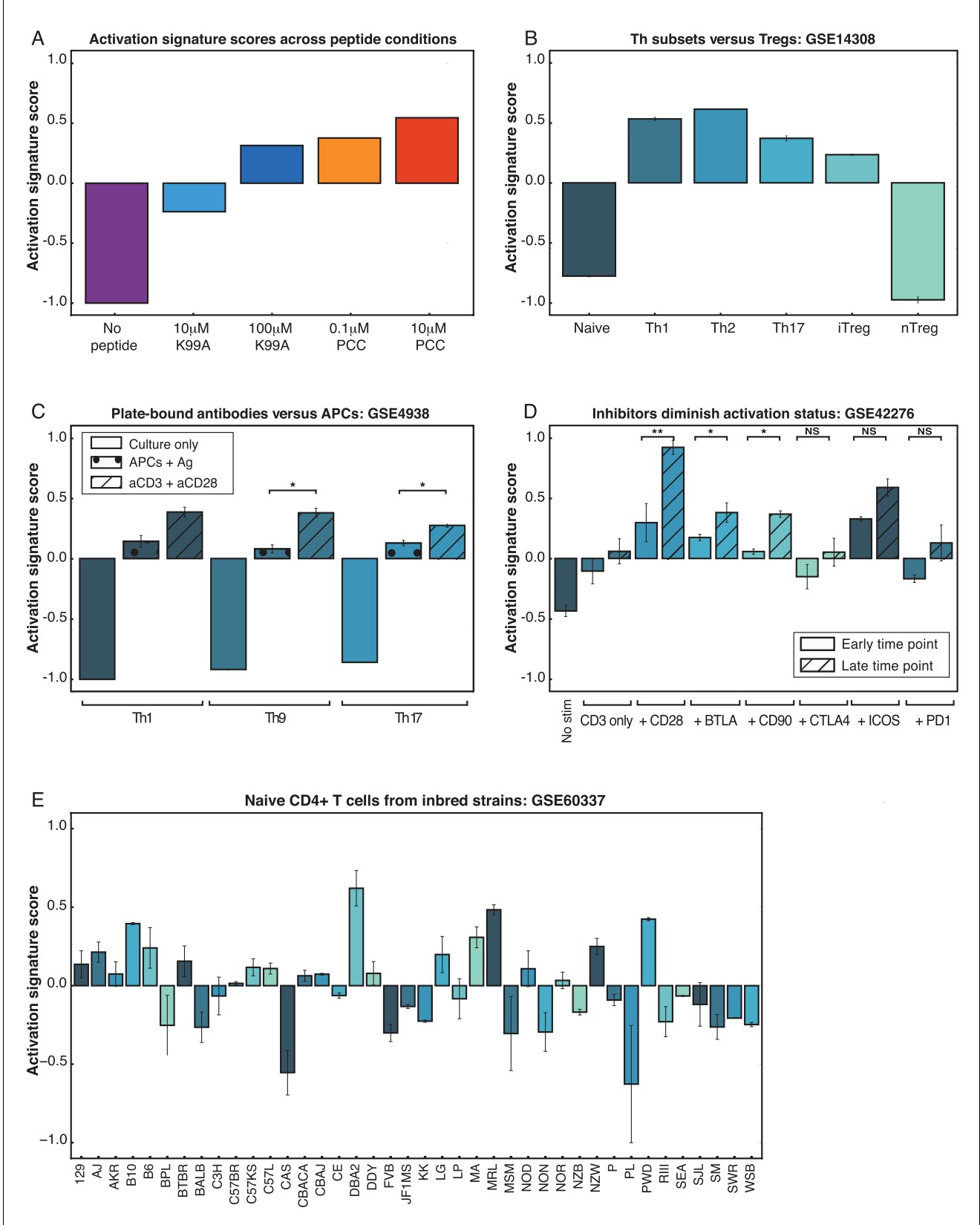

**Figure 4.** PC1 can be used to rank arbitrary CD4+ T cell data sets. (**A**) An activation score derived from the top and bottom genes along PC1 ranks the five conditions according to TCR signaling strength. The score correctly captures that 100 μM K99A and 0.1 μM PCC are very similar in activation status. *Figure 4 continued on next page*

*Figure 4 continued*

Note that the score ranks samples within an experiment, but is not an absolute metric for comparing across experiment groups. (B) The activation score can be used to compare arbitrary CD4+ T cell data sets. Here, activation scores were calculated for microarrays from helper T cell subsets, and they demonstrate that naïve cells and native regulatory T cells (nTregs) are less classically activated than Th1, Th2, and Th17 polarized subsets. (C) The activation score captures the fact that plate-bound anti-CD3 and anti-CD28 stimulation of helper T cell subsets results in stronger signaling than antigen as presented by APCs. (D) CD4+ T cells were subjected to a variety of stimulatory or inhibitory treatments: anti-CD3 alone, or anti-CD3 with anti-CD28, anti-BTLA, anti-CD80, anti-CTLA4, anti-ICOS, or anti-PD1. Gene expression profiles at early (1 hr and 4 hr) and late (20 hr and 48 hr) time points yield activation scores in line with the characterization of CD28, BTLA, CD80, and ICOS as co-stimulatory, and CTLA4 and PD1 as inhibitory. Although it might be expected that anti-CD80 would have an inhibitory effect, these results are in line with the conclusions from the originally published analysis. (E) Naïve CD4+ splenocytes were isolated from 39 mouse strains. Using the PC1-derived activation score, we can rank the CD4+ cells from each strain as either more or less activated under basal conditions. Using the activation score, we recapitulate the finding that C57Bl6 mice have more pro-inflammatory cells than BALB/c mice. The highest scoring strain, DBA/2, shows top-quartile expression of immune effectors as well as protein biosynthesis genes. (P-values based on Student's t test; *p<0.05, **p<0.01, ***p<0.001.)

The following figure supplement is available for figure 4:

**Figure supplement 1.** PC1 can be used to rank arbitrary CD4+ T cell data sets.

---

genetic perturbations of CD4+ T cell responses could be observed. For example, the activation score correctly recapitulated the findings that polarized helper subsets of CD4+ T cells were more pro-inflammatory than unstimulated cells or induced and natural regulatory T cells (*Wei et al., 2009*) (*Figure 4B*); that at the population level plate-bound anti-CD3 and anti-CD28 induced stronger activation signals than APCs plus antigen (*Tan et al., 2014*) (*Figure 4C*); that costimulation was important for achieving higher activation states but checkpoint inhibitors could block this effect (*Wakamatsu et al., 2013*) (*Figure 4D*); that knockout of Trim28, a molecule necessary for optimal IL2 production, diminished CD4+ T cell activation status (*Chikuma et al., 2012*) (*Figure 4—figure supplement 1A*); and that acute LCMV infection produced more robust activation in CD4+ T cells than chronic infection (*Doering et al., 2012*) (*Figure 4—figure supplement 1B*).

In order to test the value of the activation score, we used it to rank naïve CD4+ T cells from 39 inbred mouse strains (*Mostafavi, 2014*) (*Figure 4E*). The activation score quantified the variability in the isolated CD4+ T cells according to activation status, revealing that the genetic differences between the strains yielded different levels of activity even under unstimulated conditions. As would be predicted by known strain phenotypes, C57BL/6 cells were more activated than most strains, while BALB/c mice were less activated than most strains. The lupus-prone MRL strain and the type 1 diabetes-prone NOD strain had cells that ranked as relatively activated, whereas the type 1 diabetes-resistant NON strain had a relatively low activation score.

The strain with the highest activation score, DBA/2, had top-quartile expression of more than half of the activation signature genes (p=7.4e-30 by chi-squared test). These included a number of immune effectors such as *Irf4, Cd25, Il2rb, Nfkb1* (p105/p50), and *Nr4a1* (Nur77), as well as 17 of 32 genes from the protein biosynthesis group and 5 of 12 genes from the molecular chaperone group. Differences in the immune phenotypes of the DBA/2 strain, such as resistance to malaria, have been largely attributed to B cell-dependent mechanisms (*Bakir et al., 2006*), but the activation score here indicates that naïve CD4 T cells from DBA/2 mice are skewed toward an activated phenotype.

---

**Table 1.** Pairwise overlap between the H3K4me2 peaks. Each cell contains the count of overlapping peaks where each condition shown has at least 40 tags (normalized). In the diagonal is the total number of peaks with at least 40 tags for the given condition.

|  | No Peptide | 10 µM K99A | 100 µM K99A | 0.1 µM PCC | 10 µM PCC |
|---|---|---|---|---|---|
| No Peptide | 16012 | 14951 | 12487 | 14783 | 13105 |
| 10 µM K99A |  | 15616 | 12549 | 14838 | 13192 |
| 100 µM K99A |  |  | 12695 | 12603 | 12247 |
| 0.1 µM PCC |  |  |  | 15569 | 13281 |
| 10 µM PCC |  |  |  |  | 13444 |

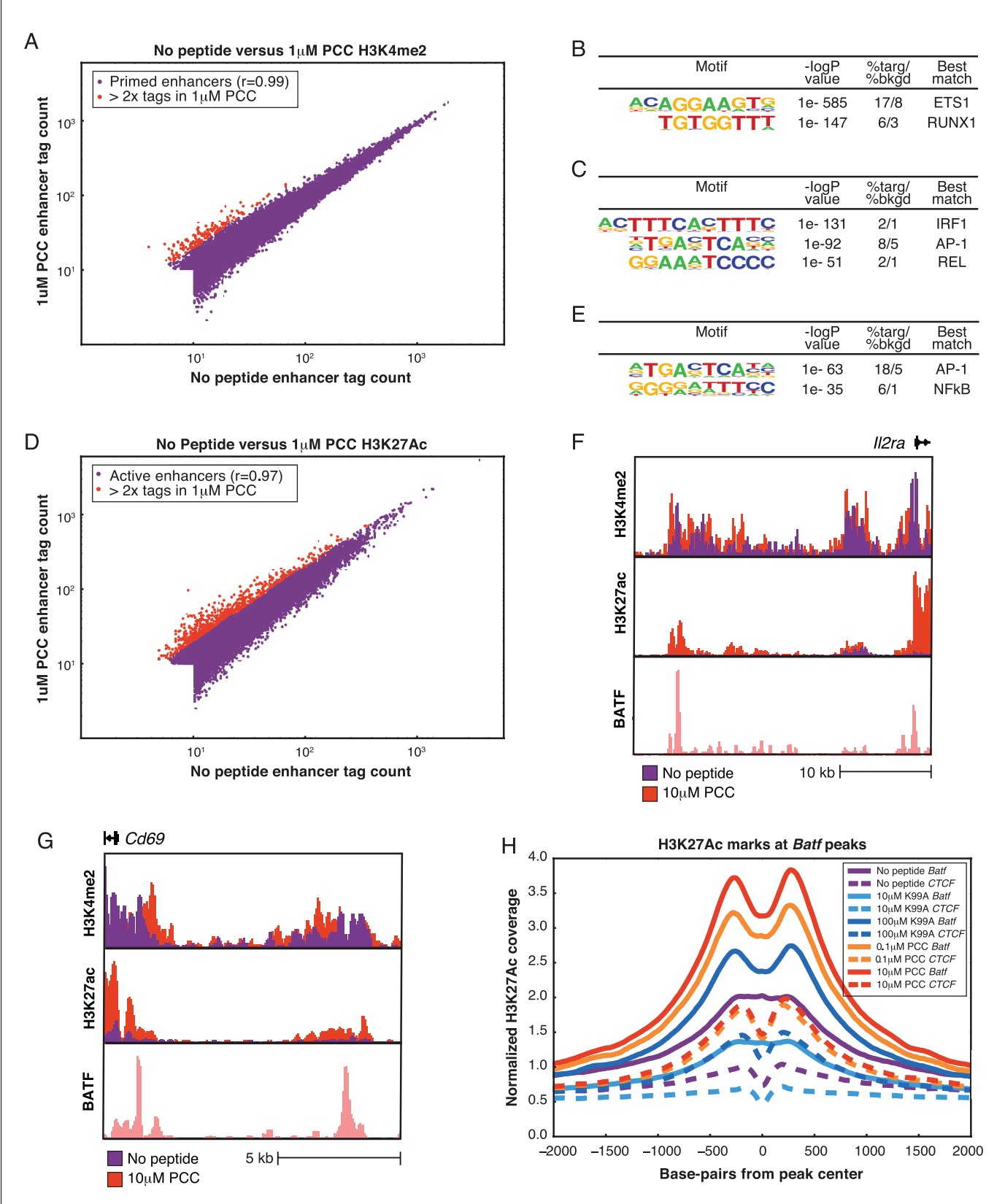

**Figure 5.** Primed enhancers are pre-existing, but gain activation markers with treatment. (**A**) Comparing primed enhancers marked by H3K4me2 peaks reveals strong correlation between untreated and treated samples. Normalized tag counts in the No Peptide condition are plotted against those in a 1
*Figure 5 continued on next page*

*Figure 5 continued*

µM PCC condition, with red dots coloring those that are more than two-fold up-regulated in the 1 µM PCC condition. The up-regulated enhancers are both few in number and low in tag count. (B) De novo motif finding identifies lineage-determining transcription factor (LDTF) motifs among primed enhancers shared by the five conditions. An ETS motif is most prominent, and a RUNX motif is likewise highly enriched over the randomly selected background. Both ETS and RUNX factors play important roles in T cell development. (C) Among primed enhancers shared by all five conditions, including the untreated condition, pro-inflammatory transcription factor motifs are enriched. An IRF family motif, AP-1 family motif (represented by BATF), and NF-κB motif (represented by REL) are all significantly enriched among shared enhancers marked by H3K4me2. (D). Comparing H3K27Ac tag counts at enhancers in No Peptide as compared to 1 µM PCC treatment reveals that many enhancers see increasing H3K27Ac deposition upon stimulation. Points in red indicate greater than two-fold increase in tags upon treatment. (E) Enhancers that are more active upon stimulation, as determined by greater than two-fold H3K27Ac tags in 1 µM PCC treatment as compared to No Peptide, are enriched for pro-inflammatory transcription factor motifs. BATF, an AP-1 family member, and NF-κB are most prominent. (F) Enhancers that are more active with stimulation are enriched near activation signature genes, as can be seen with this enhancer upstream of the activation signature gene *Il2ra* (CD25). (G) Enhancers upstream of the activation signature gene *Cd69* show an increase in H3K27Ac deposition upon treatment with 1 µM PCC. (H) Genome-wide, deposition of H3K27Ac, a marker of transcription factor activity, reflects increasing TCR signal strength at the binding sites of AP-1 family members, including BATF.

The following figure supplement is available for figure 5:

**Figure supplement 1.** Primed enhancers are pre-existing, but gain activation markers with treatment.

Three of four wild-derived strains had low activation scores: CAST, MSM, and WSB mice. All three of these wild-derived strains had bottom-quartile expression of the immune effectors *Irf1*, *Irf8*, *Stat1*, *Nfkb1*, and *Tnf*, indicating that these CD4+ T cells possess a less inflammatory gene expression profile under homeostatic conditions.

Thus, the activation score serves as a widely applicable and quantitative measure of CD4+ T cell activity, and can be used to assess the relative activation status of a variety of CD4+ T cell samples. We have developed a publically available, open source tool (see Materials and methods) to facilitate the scoring and ranking of datasets by interested parties.

## Pre-existing enhancers are leveraged to activate genes

In order to better understand the changes in genome-wide expression patterns that occurred with TCR stimulation, we compared enhancer landscapes with and without stimulation. We first performed ChIP-sequencing for H3K4me2, a marker of primed and active promoters and enhancers (*He et al., 2010*; *Kaikkonen et al., 2013*), across the five conditions. By and large, the H3K4me2-marked regions across the five conditions were very similar (*Table 1*), with a comparison of tag counts associated with specific genomic regions under no peptide or 1 µM PCC illustrated in *Figure 5A*. Though there were some enhancers showing at least two-fold change in H3K4me2 tag counts across conditions, these regions were at the lower end of the tag count range and therefore differences were not significant (*Figure 5A*, red points). Thus, the gene expression and phenotypic changes seen after activation were not due to selection of new signal-dependent enhancers.

At promoters, H3K4me2 marks were shared across the five conditions, but activation signature genes showed spreading of the H3K4me2 mark along the body of the gene after TCR stimulation. In contrast, H3K4me2 peaks were narrow and focal for the untreated condition at many of these genes. This effect can be seen at *Cd69* (*Figure 5—figure supplement 1A*) and *Irf4* (*Figure 5—figure supplement 1B*), resulting in a global increase of the ratio of gene body tags to promoter tags at activation signature genes (*Figure 5—figure supplement 1C*) but not genes in the bottom 10% of PC1 (*Figure 5—figure supplement 1D*). This implies that the process of activating these genes subsequent to TCR stimulation induces deposition of the dimethyl mark along the body of the genes as they are transcribed.

## Motif analysis reveals lineage-determining and signal-dependent TFs

We used de novo motif finding (*Heinz et al., 2010*) to identify lineage-determining transcription factors (LDTFs), also known as pioneer factors or master regulators, which establish cell-type-specific enhancer landscapes, and determine the available open chromatin for subsequent binding of signal-dependent transcription factors (SDTFs) (*Garber et al., 2012*; *Heinz et al., 2013*; *Mullen et al., 2011*; *Soufi et al., 2012*; *Trompouki et al., 2011*). The top motif was an ETS motif (*Figure 5B*), capable of being bound by a number of ETS factors that are expressed in CD4+ T cells, including

**Table 2.** Pairwise overlap between the H3K27Ac peaks. Each cell contains the count of overlapping peaks where each condition shown has at least 40 tags (normalized). In the diagonal is the total number of peaks with at least 40 tags for the given condition.

|  | No Peptide | 10 µM K99A | 100 µM K99A | 0.1 µM PCC | 10 µM PCC |
|---|---|---|---|---|---|
| No Peptide | 6410 | 3591 | 5791 | 5988 | 5977 |
| 10 µM K99A |  | 3671 | 3557 | 3594 | 3584 |
| 100 µM K99A |  |  | 8877 | 8699 | 8789 |
| 0.1 µM PCC |  |  |  | 9867 | 9368 |
| 10 µM PCC |  |  |  |  | 10021 |

Ets1, Ets2, and Elf1 (*Anderson et al., 1999*). These enhancers tend to be shared across similar cells as well as thymic T cell precursors (*Heinz et al., 2010*; *Zhang et al., 2012*). Similarly, Runx factors play an important role in T cell development (*Wong et al., 2011*), and correspondingly the Runx family motif was highly enriched among primed enhancers.

Several known motifs for SDTFs were also enriched among the H3K4me2-marked enhancers (*Figure 5C*), including an Interferon Regulatory Factor (IRF) motif. Although IRFs respond to interferon signaling (*Ozato et al., 2007*), and would not be expected to be active in unstimulated cells (*Murphy et al., 2013*; *Glasmacher, 2012*; *Li et al., 2012*), it is possible that the IRF motif is a 'memory' of states of activation during the development of CD4+ T cells, and indeed IRF motifs can be found in several related cell types and multiple stages of thymocyte development (*Figure 5—figure supplement 1E*), suggesting that the primed enhancers in naïve CD4+ T cells are predisposed to act as binding sites for key SDTFs (*Heinz et al., 2010*; *Zhang et al., 2012*; *Buecker et al., 2014*; *Mikkelsen et al., 2010*; *Mishra et al., 2014*; *Vahedi et al., 2012*). Similarly, an AP-1 motif and an NF-κB motif were significantly enriched in primed enhancers (*Figure 5C*), corresponding with the fact that TCR signaling greatly increases activity of both of these transcription factors (*Huang and Wange, 2004*; *Rincón and Flavell, 1994*).

Given that H3K4me2-marked regions were not substantially changed across the five conditions, we next performed ChIP-sequencing for H3K27Ac, a marker for active enhancers (*Creyghton et al., 2010*), under a stimulated condition (1 µM PCC) and the unstimulated condition. In contrast to H3K4me2, a substantial portion of enhancers exhibited increases in the H3K27Ac activation mark (*Figure 5D*, *Table 2*). The union set of enhancers was enriched for a similar set of motifs as the primed enhancers (*Figure 5—figure supplement 1F,G*). Enhancers that became more active with TCR engagement were highly enriched for both AP-1 and NF-κB motifs (*Figure 5E*). Further, this group of enhancers was more likely to be proximal to activation signature genes than would be expected at random (p-value = 2.0e-20 by chi-squared test). These enhancers included, for example, those upstream of *Il2ra* (CD25; *Figure 5F*) and *Cd69* (*Figure 5G*).

To investigate whether there was a quantitative relationship between TCR signal strength and enhancer activation, we performed independent ChIP-Seq for H3K27Ac in response to both peptides at low and high concentrations. Given the prevalence of the AP-1 motif in the signal-responsive enhancers, we analyzed H3K27Ac tag counts at AP-1 binding sites genome-wide using publicly available ChIP-Sequencing data from in vitro activated TH17 cells (*Li et al., 2012*). There was an increase in H3K27Ac deposition at AP-1 binding sites that reflected the graded strength of TCR signaling (BATF shown in *Figure 4H*; other AP-1 family members in *Figure 5—figure supplement 1H,I*), indicating that AP-1 binding sites became more active in a graded manner corresponding to increasing TCR signaling. Graded changes in H3K27Ac were much less pronounced at CTCF binding sites, which occur at enhancers but are also more broadly distributed and play roles in establishing boundary elements.

## Super-enhancers prime signaling genes

We next looked at changes in super-enhancers (*Hnisz et al., 2013*; *Lovén et al., 2013*; *Whyte et al., 2013*) upon activation using the H3K27Ac mark. Most super-enhancers (approximately 450 out of 700 total) identified were shared by both the unstimulated and stimulated conditions. GO analysis of genes nearby the shared super-enhancers showed enrichment for leukocyte activation

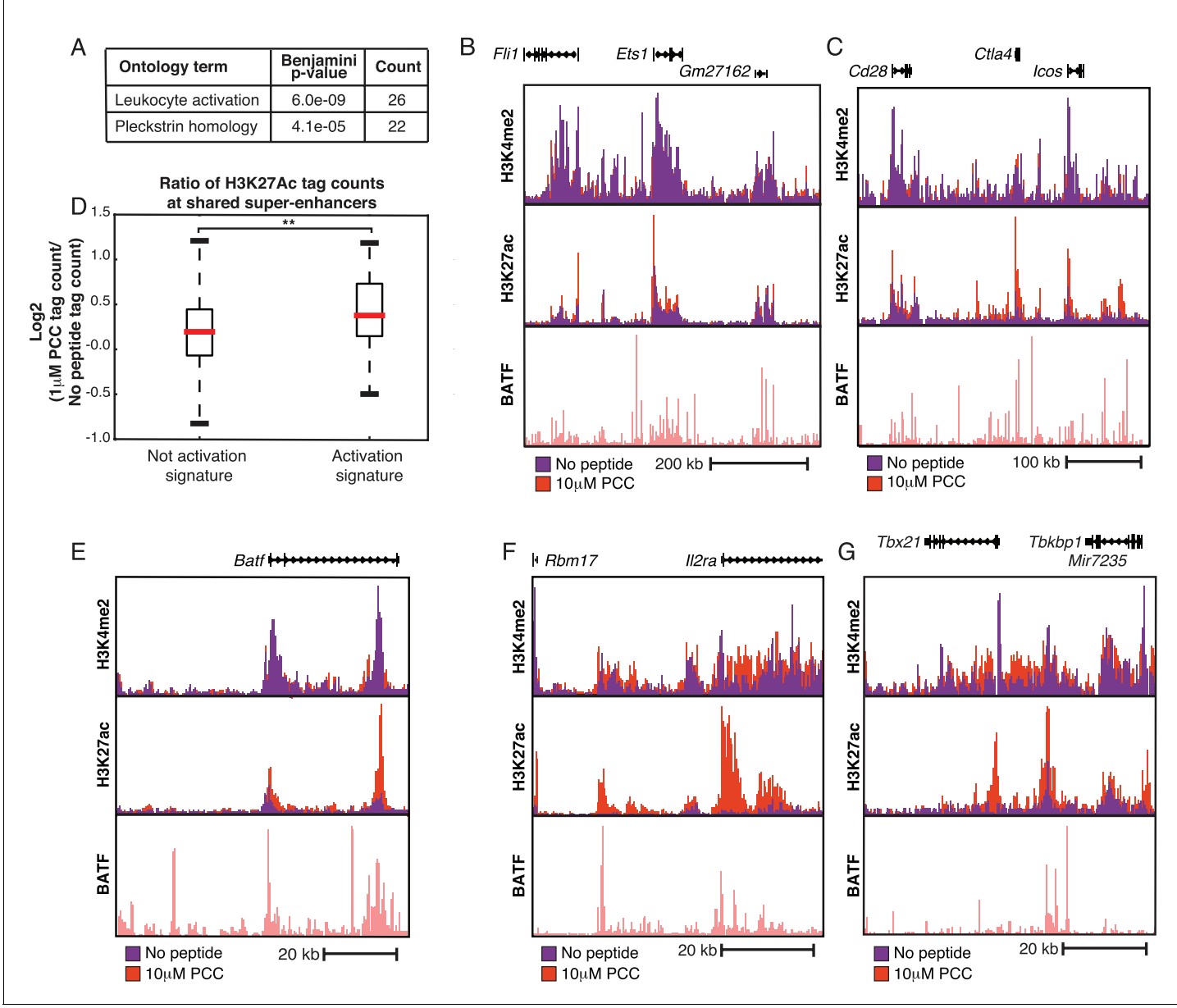

**Figure 6.** Super-enhancers prime T cell activation genes. (**A**) Gene Ontology (GO) analysis of genes nearest to the ~450 super-enhancers shared by treated and untreated conditions show enrichment for T cell activation genes. P-values shown are Benjamini-Hochberg adjusted p-values. (**B**) H3K27Ac marks a large super-enhancer around the lineage-determining transcription factor *Ets1* in both the No Peptide and 1 µM PCC conditions. The super-enhancer spans the ~600 kbp region shown. (**C**) The ~400 kbp super-enhancer region encompassing *Cd28, Ctla4*, and *Icos* is marked by H3K27Ac in both treated and untreated conditions. (**D**) Despite being heavily marked by H3K27Ac in both untreated and treated conditions, shared super-enhancers near activation signature genes show a significant gain in H3K27Ac tags in response to stimulation as compared to the shared super-enhancers not proximal to activation signature genes. In other words, basally primed super-enhancers near activation signature genes see significant increases in activity upon stimulation, correlating with increased gene expression at the activation signature genes. (**E**) Some regions of H3K27Ac deposition required TCR stimulation to pass the super-enhancer threshold, as can be seen here at the ~60 kbp region encompassing BATF, an AP-1 family member. While H3K27Ac is clearly present under basal conditions, there is a substantial increase in enhancer activity upon treatment with 10 µM PCC. (**F**) *Il2ra* (CD25) shows increased enhancer activity and formation of a super-enhancer in the treated condition. (**G**) Similarly, the region surrounding *Tbx21* (*Tbet*) shows substantial increases in activity subsequent to stimulation, resulting in the formation of a super-enhancer. (p-values based on Student's t test; *p<0.05, **p<0.01, ***p<0.001).

The following figure supplement is available for figure 6:

**Figure supplement 1.** Super-enhancers prime T cell activation genes.

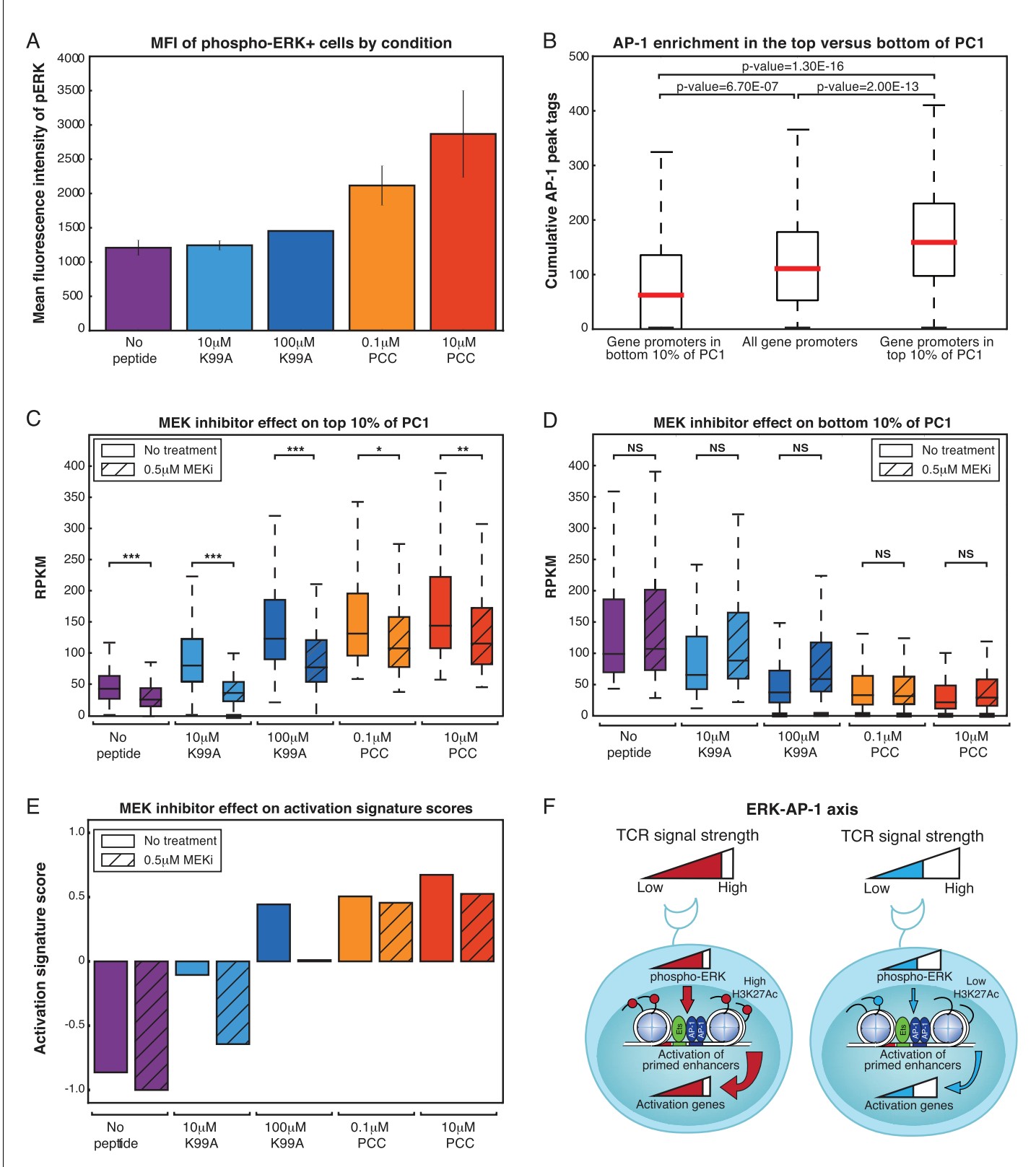

**Figure 7.** ERK signaling translates TCR signal strength into graded gene expression. (**A**) ERK phosphorylation is a measure of ERK pathway activity. Flow cytometry for phospho-ERK after 3.5 hr of co-culturing shows that, on a per-cell basis, increasing signal strength yields increasing levels of phospho-ERK among CD4+ phospho-ERK+ cells. (**B**) ERK pathway activation is upstream of the transcription factor AP-1. ChIP-sequencing tags for four

*Figure 7 continued on next page*

*Figure 7 continued*

AP-1 family members (BATF, cJun, JunB, and JunD) in Th17 cells shows that there is an enrichment for AP-1 binding near the promoters (plus or minus 1,000 bp from the TSS) of activation signature genes (top 10% of PC1) as compared to all genes or the genes in the bottom 10% of PC1. (C) A MEK inhibitor that dampens signaling upstream of the ERK pathway preferentially diminishes expression of activation signature genes, as seen in the fact that the RPKM of genes in the top 10% of PC1 is significantly reduced with treatment. (D) The reduction of RPKM seen with the activation signature genes is not a general effect, as the RPKM of the genes in the bottom 10% of PC1 are not significantly affected by MEK inhibitor treatment. (E) We quantified the effect of MEK inhibitor treatment using the activation signature score. Treatment with the MEK inhibitor reduces the activation signature score for all samples. (F) Schematic of the ERK-AP-1 axis. See text for details. (p-values based on Student's t test; *p<0.05, **p<0.01, ***p<0.001).

The following figure supplements are available for figure 7:

**Figure supplement 1.** ERK signaling translates TCR signal strength into graded gene expression.

**Figure supplement 2.** AP-1 enrichment is independent of expression level.

**Figure supplement 3.** AP-1 enrichment is independent of expression level.

genes as well as Pleckstrin homology genes, which are critical components of a number of kinase signaling pathways downstream of the TCR (*Figure 6A*), indicating that super-enhancers in CD4+ T cells prime genes important for inflammatory signaling. These basally-primed super-enhancers included regions near key T cell genes such as *Ets1* (*Figure 6B*), *Runx1* (*Figure 6—figure supplement 1A*), *Ctla4*/Icos/CD28 (*Figure 6C*), and *Irf4* (*Figure 6—figure supplement 1B*). Notably, even though many of the super-enhancers exist prior to stimulation, super-enhancers near activation signature genes show an increase in H3K27Ac signal subsequent to TCR signaling (*Figure 6D*).

118 of the 568 super-enhancers identified after TCR stimulation were not identified as super-enhancers in the unstimulated condition. The super-enhancers that required TCR signaling were enriched for leukocyte activation genes (Benjamini-Hochberg adjusted p-value = 4.6e-7) crucial for T cell activation, including *Batf* (*Figure 6E*), *Il2ra* (CD25, *Figure 6F*), *Tbx21* (Tbet, *Figure 6G*), *Lag3* (*Figure 6—figure supplement 1C*), and *Stat5b* (*Figure 6—figure supplement 1D*).

## ERK translates TCR signal strength downstream

The Ras/Raf/Mek/Erk pathway downstream of the TCR activates the AP-1 transcription factor family through a series of phosphorylation events and transcriptional induction of immediate-early genes (*Murphy et al., 2013*; *Rincón and Flavell, 1994*). Given the fact that the AP-1 motif was enriched at enhancers showing increasing activity and the fact that AP-1 binding sites saw increasing H3K27Ac deposition, we sought to determine whether AP-1 and the ERK pathway were relevant to the increasing expression of activation signature genes across the conditions. We first compared the level of phosphorylated ERK (p-ERK) in each condition using flow cytometry, and found that, like CD69 and CD25, the amount of p-ERK in the p-ERK+ cells varied on a per-cell basis in each condition, increasing with TCR signal strength (*Figure 7A*). Frequency of pERK+ cells also differed across peptide conditions (*Figure 7—figure supplement 1A*), indicating that both frequency and single-cell response level play a role in the effect of MEK inhibitors. As the increasing levels of p-ERK paralleled the general pattern of expression of the activation signature genes, we assessed binding frequencies of AP-1 factors in the gene promoters of the activation signature genes as compared to the bottom 10% of PC1 genes, and found that activation signature genes showed a significantly higher frequency of AP-1 binding (*Figure 7B*). This enrichment for AP-1 binding was not dependent on the expression level of the genes in each group, as segmenting the genes by RPKM showed the same pattern (*Figure 7—figure supplement 2*).

ERK pathway activation and AP-1 binding therefore seemed to correlate well with the graded profile of activation signature genes and the increasing activation score across the samples. In order to determine whether the gradual increase in ERK pathway activation was causal in translating TCR signaling into gradual increases in the expression of activation signature genes, we pretreated the CD4+ T cells with a low-dose MEK inhibitor (MEKi). MEK inhibition upstream of ERK was capable of suppressing p-ERK activity entirely, and titration of MEKi yielded intermediate levels of p-ERK on a per-cell basis (*Figure 7—figure supplement 1B*).

Low-dose MEK inhibition decreased the levels of the extracellular signaling marker CD69 (*Figure 7—figure supplement 1C*), which was in the top 10% of PC1, but this suppression was not universal, as CD4, an example of a gene not in the top 10% of PC1, was not significantly affected (*Figure 7—figure supplement 1D*). In order to see if this preferential suppression of activation signature genes was widespread, we performed RNA-seq on the five conditions after pretreatment with MEKi at IC50 (0.5 μM). If TCR signaling strength upstream of pERK yields graded levels of ERK that are in turn essential for the graded levels of activation response genes, then reduction of pERK levels should move each sample downwards in activation score, such that the high-dose, high-affinity case looks like the low-dose, high-affinity; the low-dose, high-affinity looks like the high-dose, low-affinity; and so on.

Accordingly, MEK inhibition at IC50 decreased expression of activation signature genes (*Figure 7C*, *Figure 7—figure supplement 1E*), but, as with extracellular expression of CD69, this effect was selective; expression of genes in the bottom 10% was increased or unchanged (*Figure 7D*, *Figure 7—figure supplement 1F*). The decrease in expression among genes in the top 10% of PC1 upon MEKi treatment was not dependent on expression level of the genes, as the effect was consistent when genes were segmented by RPKM level (*Figure 7—figure supplement 3*). Using the activation score to assess total T cell activation status, we found that MEKi shifted each sample down in score (*Figure 7E*), as would be expected if the graded levels of pERK seen with each condition were prescriptive of the activation status of the condition. Thus, graded levels of pERK downstream of the TCR help to translate the analog activation signal to graded levels of enhancer activity and gene expression genome-wide (*Figure 7F*).

## Discussion

Understanding how CD4+ T cells respond to ligands of different doses and affinities is critical to understanding the nature of the adaptive immune response to both pathogens and self. Here, we have shown that the traditional model of a purely digital TCR response is too simple; on a per-cell basis, stronger TCR signals result in higher levels of phosphorylated ERK, a proportional increase in enhancer acetylation, and quantitative increases in activation markers such as CD69 and CD25 (*Figure 7F*). As a result of both these single-cell differences and the increasing frequency of respondent cells, varying the dose or the affinity of the pMHC-TCR interaction results in a gene expression profile that is graded corresponding to increasing strength of TCR signaling. We observed no evidence for Th1/Th2 skewing as a function of peptide dose/affinity, which may reflect the specific peptides used for stimulation. Notably, the predominance of PC1 and the graded gene expression patterns together indicate that dose and affinity are not interpreted separately downstream of the TCR, but rather overall signaling strength sets the level of activation across the population. A similar conclusion has been drawn by looking at the sum of activation parameters induced by antigen-, co-stimulatory- and cytokine-receptors (*Marchingo, 2014*).

Prior analysis of T cell responses at the single cell level showed that single T cell engagement of different numbers of identical peptide agonists (titration of signal strength) resulted in digital responses read out as increasing numbers of T cells producing the same amount of TNF or IL2 protein (*Huang et al., 2013*). Intriguingly, different levels of responses were seen when naïve cells were compared with blast and memory cell responses, such that the rate of TNF and IL2 synthesis in blasts and memory cells was nearly ten-fold higher than in naïve cells. Our findings are consistent with these earlier studies in that TNF and IL2 are members of the 'digital' class of RNAs identified in CD69+ cells (*Figure 3—figure supplement 1E*). Furthermore, the increase in expression of genes involved in protein expression revealed by the present studies could at least partly explain the observation that the rates of TNF and IL2 protein production are increased in blasts and memory cells in comparison to naïve cells. The analogue increase in protein biosynthetic machinery could also at least partly explain the observation that CD69 expression is digital at the level of mRNA but analogue at the level of protein expression (*Figure 1A*, *Figure 3—figure supplement 1F*).

Ranking genes along a primary axis of variation allowed us to extract a set of activation signature genes that increase in a graded fashion at the population level proportionally to TCR signal strength, and further to establish an activation score that can rank arbitrary CD4+ T cell samples by the strength of signaling. The data presented here therefore gives us a greater understanding of the

CD4+ T cell response to ligands of varying concentrations and affinities, and informs our understanding of CD4+ T cell activation under diverse conditions.

Significant differences in primed enhancers have been demonstrated under several stimulating conditions in macrophages, and demonstrate the ability of cells to quickly remodel chromatin to initiate particular gene expression programs (*Kaikkonen et al., 2013*; *Ostuni et al., 2013*). Surprisingly, we did not find significant changes in the primed enhancer landscape upon TCR activation in CD4+ T cells. Similarly, even the more labile activation mark H3K27Ac was largely similar across conditions, with many activation genes marked as super-enhancers even before stimulation. While it remains to be seen whether non-TCR signaling pathways or polarizing conditions induce more dramatic changes, the data presented here indicates that the CD4+ T cell enhancer landscape is largely pre-established, with subsets of H3K4me2-marked enhancers increasing in activity, but little in the way of de novo enhancer establishment. This finding helps to explain the speed and plasticity of the CD4+ T cell response (*Zhou et al., 2009*)—if all enhancers are primed basally, and many are even activated basally, then pro-inflammatory transcription factors can bind at established enhancers and initiate new gene expression programs with minimal additional transcriptional machinery.

Both the frequency of AP-1 binding and the level of pERK correlate with the strength of TCR signaling and the graded expression of activation signature genes. At least one of the feedback loops leading to digital TCR signaling, the son of sevenless (SOS) positive feedback loop, exists upstream of ERK, and it has been shown in thymocytes that pERK signaling is digital (*Das et al., 2009*; *Daniels et al., 2006*; *Prasad et al., 2009*). However, studies focused on EGFR signaling upstream of ERK indicated that discrete pulses of ERK activity result in quantitative levels of downstream signaling activity (*Albeck et al., 2013*). Our analog results for pERK can be interpreted to support the notion that TCR signaling in thymocytes functions differently than TCR signaling in mature T cells. Notably, both Themis and SOS, two key components of digital signaling in thymocytes, do not seem to be critical to mature T cell signaling (*Fu et al., 2013*; *Warnecke et al., 2012*).

The graded levels of pERK in CD4+ T cells prove important for downstream enhancer and gene activity. We have here established a mechanistic link between the level of ERK signaling and the expression patterns of activation signature genes by using an inhibitor of MEK, upstream of ERK. Low-dose MEK inhibition selectively decreased expression of the activation signature genes such that the activation score under the inhibited conditions was incrementally decreased. This indicates that the analog levels of pERK seen on a per-cell basis are translated at a population level into increased enhancer and gene activity, and that 'turning down' pERK levels selectively diminishes the activation status of the cells. This finding is of particular interest in light of the clinical availability of numerous RAF, MEK, and ERK inhibitors (*Zhao and Adjei, 2014*; *Samatar and Poulikakos, 2014*). Our findings suggest that low-dose ERK pathway inhibition could be used to selectively decrease the activity of activation signature genes in CD4+ T cells, achieving low-level immunosuppression without killing T cells or completely removing their ability to respond to TCR signaling. Further, the effect of MEK inhibitors on CD4+ T cells raises questions about the immunosuppressive effects of using MEK inhibitors in cancer treatment, especially as current clinical trials combine MEK inhibitors with checkpoint-blockade inhibitors (*Zhao and Adjei, 2014*; *Vella et al., 2014*).

Notably, NFκB is one of many transcription factors known to play important roles in T cell biology, and indeed we find a κB motif enriched among enhancers that are responsive to stimulation (*Figure 5E*). Further research as to the relationship between the strength of TCR signaling and other signal-dependent transcription factors such as NF-κB and NFAT is warranted.

In sum, this study makes use of a unique model system to dissect the transcriptional responses of CD4+ T cells to increasing strength of signaling, and demonstrates that analog levels of pERK within the context of digital TCR signaling flow downstream to result in graded gene expression profiles and enhancer landscapes that can be used to characterize CD4+ T cell signaling at large.

# Materials and methods

## Mice

AND mice on a B10.BR background were received from Dr. Michael Croft (*Rogers and Peptide dose, 1999*; *Rogers et al., 1998*) and bred in a specific pathogen free facility. All animal

experiments were in compliance with the ethical standards set forth by UC San Diego's Institutional Annual Care and Use Committee (IUCAC).

## Cells

Spleens were extracted and manually digested. CD11c+ cells were isolated using Miltenyi Biotec Inc. (San Diego, CA) MACS magnetic cell separation with positive selection for CD11c (CD11c, Biolegend, cat. no. 117304). Subsequently, the CD11c- splenic fraction was used to negatively select for naïve CD4+ T cells using the Miltenyi MACS system with the following antibodies: CD11c (Biolegend, cat. no. 117304); CD45R (eBioscience, cat. no. 13-0452-86); CD11b (eBioscience, cat. no. 13-0112-86); CD25 (eBioscience, cat. no. 36-0251-85); CD49b (eBioscience, cat. no. 13-5971-85); CD69 (eBioscience, cat. no. 13-0691-85); CD8a (eBioscience, cat. no. 13-0081-86); Ly-6G (eBioscience, cat. no. 13-5931-86); MHC class II (eBioscience, cat. no. 13-5321-85); TER-119 (eBioscience, cat. no. 13-5921-85). CD11c+ and CD4+ cells were cultured at a ratio of 1:2 in 96-well round-bottom plates for 24 hr, 108 hr (for proliferation assay), or 3.5 hr (for ERK phosphorylation staining). Peptides were added at indicated concentrations with the CD11c+ and CD4+ cells in DMEM supplemented with 10% Fetal Bovine Serum. For sequencing experiments, CD4+ cells were re-isolated from the culture using the Miltenyi MACS system and the same set of antibodies as above less CD25 and CD69. For phospho-ERK staining, whole splenic cells were used, rather than purified CD11c+ and CD4+ cells.

## Peptides

Peptides were ordered from Peptide 2.0 (Chantilly, VA) with the following amino acid sequences (*Rogers and Peptide dose, 1999*; *Rogers et al., 1998*):

    PCC – KAERADLIAYLKQATAK
    K99A – KAERADLIAYLAQATAK
    Y97K – ANERADLIAKLKQATK
    K99E – ANERADLIAYLEQATK
    MCC – ANERADLIAYLKQATK

Lyophilized peptides were resuspended in water, and added at the indicated concentrations to the cell cultures. Unstimulated CD4+ cells received an equivalent amount of water alone.

## Flow cytometry

Flow cytometry was performed on a LSR II and LSR Fortessa, both from BD Biosciences (San Jose, CA). Cells were stained as per manufacturers' protocols with the following antibodies: CD4-APC (eBioscience, cat. no. 17-0042-83); CD4-PE-Cyanine7 (BioLegend, cat. no. 116016); CD69-FITC (eBioscience, cat. no. 11-0691-82); CD25-PE (eBioscience, cat. no. 12-0251-82); Valpha11-FITC (BD Pharmingen, cat. no. 553222); Vbeta3-PE (BD Pharmingen, cat. no. 553209); CD11c-PE-Cyanine7 (eBioscience, cat. no. 25-0114-82); IRF4-PerCP-Cy5 (eBioscience, cat. no. 46-9858-80); Tbet-PE (Santa Cruz Biotechnology, cat. no. SC-21749); CD122-PE (BioLegend, cat. no. 105905); Ly6a-APC-Cyanine7 (BioLegend, cat. no. 108125); CD200-APC (BioLegend, cat. no. 123809); TNFSF11-APC (BD Biosciences, cat. no. 560296); phospho-ERK-Alexa Fluor 488 (Cell Signaling, cat. no. 4344S). Live/dead staining was performed using Fixable Aqua (Life Technologies, cat. no. L34957; or Biolegend, cat. no. 423102). Cells were gated on CD4+, Aqua- cells. For phosphor-ERK staining, permeabilization was performed using BD Phosflow Perm Buffer III (cat. no. 558050) and BD Fix Buffer I (cat. no. 557870). Analysis was performed with FlowJo v10.6 (Tree Star; Ashland, OR). All flow cytometry results shown are representative of at least two biological replicates, and the results shown in *Figure 1* were reflected with samples held out of each high-throughput sequencing assay.

## mRNA flow cytometry

CD11c+ cells and naïve CD4+T cells were isolated form spleens of AND mice using magnetic separation as described above. CD11c+ cells and CD4+ T cells were cultured at a ratio of 1:2 in 96 –well round bottom plates with various concentrations of K99A (10 μM and 100 μM) and PCC (0.1 μM and 10 μM).

After 22 hr, cells were harvested and mRNA expression of *Irf4* was analyzed on single cell level by flow cytometry in combination with CD4, and CD69 protein staining, using FlowRNA II Assay kit (Affymetrix eBioscience) according to manufacturer's protocols (*Porichis et al., 2014*). Cells were

analyzed with a BD AriaII flow cytometer. Data were analyzed using FlowJo v887software (Tree Star). Two technical and two biological replicates were obtained across all conditions using the two different peptides at two different concentrations to stimulate naïve CD4+ T cells.

## Sequencing

Prior to sequencing, CD4+ T cells were separated from the co-cultured cells using Miltenyi MACS negative selection as described above for the initial culturing. ChIP-sequencing for H3K4me2 and H3K27Ac in the 1 μM peptide treatments was performed as described (*Gilfillan et al., 2012*), with the following modifications: sodium butyrate was used to inhibit de-acetylation; and three RIPA and three LiCl washes were performed instead of five and one. ChIP-sequencing for H3K4me2 and H3K27Ac across the five conditions was performed as described (*Gosselin et al., 2014*). RNA-sequencing was performed as described, with minor modifications (*Wang et al., 2011*). Two replicates of the H3K4me2 ChIP-seq across all five conditions were obtained; three replicates across three conditions and one across all five conditions were obtained for the H3K27Ac ChIP-seq; and two replicates across all five conditions of the RNA-seq were obtained.

ChIP-sequencing antibodies used were: H3K4me2 (Millipore, cat. no. 07–030) and H3K27Ac (Abcam, cat. no. ab4729 and Active Motif, cat. no. 39135).

To analyze the transcriptome of activated CD4+ T cells, we sorted CD69 positive cells. CD11c+ and CD4+ T cells were cultured for 22 hr as described above using two peptides at different concentrations K99A (10 μM and 100 μM) and PCC (0.1 μM and 10 μM). After harvesting, cells were stained with Zombie Aqua live/dead stain (Biolegend) and with CD4-PE (clone RM4-5, eBioscience) and CD69-PE Cy7 (clone H1.2F3, eBioscience) conjugated antibodies. Cells were sorted with a BD AriaII cell sorter using a 70 μm nozzle. Live CD4+ cells were sorted into two populations according to the expression of the CD69 activation marker. CD69+ cells from the various culture conditions were used for RNA extraction. Unstimulated CD69- cells were used as controls. After extraction with Trizol, RNA was PolyA-selected (MicroPoly(A) Purist kit, Ambion) and libraries were generated as previously described (*Heinz et al., 2013*). Samples were sequenced using NextSeq2 (Illumina) according to manufacturer recommended protocols.

## MEK inhibitor treatment

CD4+ T cells, isolated as described above, were pre-treated with 0.5 μM Promega U0126 (cat. no. V1121) for thirty minutes at 37°C. CD11c+ cells and peptides were subsequently as indicated and cultured in the presence of the inhibitor for 24 hr.

## Analysis

ChIP-sequencing reads were mapped to the mm10 genome using Bowtie2 (*Langmead and Salzberg, 2012*), and RNA-sequencing reads were mapped using STAR (*Dobin et al., 2013*). Default allowed error rates were used, and only uniquely mapping reads were used in downstream analysis. Initial processing of aligned data and peak calling was performed using Homer (*Heinz et al., 2010*). IDR analysis (*Landt et al., 2012*) for ChIP-sequencing replicates was performed using the homer-idr package (*Allison, 2015*). Vespucci (*Allison et al., 2014*) was used for counting AP-1 tags in gene regions.

Gene Ontology analysis was performed using the Metascape Gene Annotation and Analysis Resource tool (http://metascape.org/gp/index.html#/main/step1) using the express analysis settings (*Tripathi et al., 2015*).

Super-enhancers were called using Homer (*Heinz et al., 2010*), which follows the published procedure (*Hnisz et al., 2013*; *Lovén et al., 2013*; *Whyte et al., 2013*) by first stitching together peaks into larger regions and then sorting regions by normalized H3K27Ac tag count. Region scores are plotted against rank, and a threshold is defined by finding the point at which the tangent to the plotted rank-scores is one. Regions past that threshold are called super-enhancers.

Underlying data sets, including RPKM values and peaks, as well as code for all analyses described is publicly available at https://github.com/karmel/and-tcr-affinity. Analyses were performed using iPython Notebook (*Perez and IPython, 2007*). Clustering and PCA was performed using the scikit-learn package (*Pedregosa, 2011*). For full execution details and parameters, please see the code in the Github repository linked here.

## Activation signature scores

To generate the list of genes used in the activation signature score, we separately ran Principal Component Analysis on two replicates of RNA-Seq data and also the combined expression data from both replicates. Genes with an RPKM less than 100 in the No Peptide condition or a standard deviation greater than 20% of the No Peptide expression level across replicates were then omitted from the target set of genes. Remaining genes were sorted along PC1, and genes that were in the top ten percent in all three data sets (215) or the bottom ten percent in all three data sets (137) were included in the set of activation signature score genes used in analysis.

To compute the activation signature score, we take the dot product of the values of genes along PC1 in the combined RNA-Seq data set and the mean-centered expression levels for those genes for each sample in an experimental data set, yielding a single scalar score for each experiment. The scores across samples are then scaled by the max score, ensuring values are in the range of [-1, 1].

The activation signature score tool is described and downloadable here: https://github.com/karmel/and-tcr-affinity/tree/master/andtcr/rna/activationscore.

## Public data

Publicly available datasets used for the analyses in Figure 3 and Figure 3—figure supplement 3 is available from GEO with the following Accession Codes: GSE14308, GSE32224, GSE41866, GSE42276, GSE54938, and GSE60337. AP-1 binding data is from GSE39756. For *Figure 5E*, the following datasets were used: GSE56456, GSE31233, GSE40463, GSE21365, GSE56098, GSE21512.

## Accession codes

Raw and processed data are provided in the Gene Expression Omnibus (GEO) under accession number GSE69545.

## Acknowledgements

The authors would like to thank Leslie Van Ael and David Allison for assistance with preparation of the manuscript. These studies were primarily supported by NIH grants DK091183, CA17390, DK063491, R01-AI103440, and the San Diego Center for Systems Biology (GM085764). KAA was supported by F31-AI112269 (NIAID); ELS was supported by K01-DK095008 (NIDDK); TDT was supported by T32-CA009523 (NIH); and DG was supported by a Canadian Institutes of Health Research Fellowship. The authors declare no conflict of interest.

## Additional information

### Competing interests

CKG: Reviewing editor, *eLife*. The other authors declare that no competing interests exist.

### Funding

| Funder | Grant reference number | Author |
|---|---|---|
| National Institute of Allergy and Infectious Diseases | F31-AI112269 | Karmel A Allison |
| National Institutes of Health | RO1AI073885 | Jana G Collier |
| Canadian Institutes of Health Research | | David Gosselin |
| National Institutes of Health | T32-CA009523 | Ty Dale Troutman |
| National Institutes of Health | K01-DK095008 | Erica L Stone |
| National Institutes of Health | CA173903 | Christopher K Glass |
| National Institutes of Health | DK091183 | Christopher K Glass |
| National Institutes of Health | DK063491 | Christopher K Glass |
| National Institutes of Health | GM085764 | Christopher K Glass |

| National Institutes of Health | CA17390 | Christopher K Glass |
| National Institutes of Health | R01-AI103440 | Christopher K Glass |

The funders had no role in study design, data collection and interpretation, or the decision to submit the work for publication.

### Author contributions
KAA, Designed the studies, Performed experiments and analyzed data, Conception and design, Acquisition of data, Analysis and interpretation of data, Drafting or revising the article; ES, Performed experiments, Conception and design, Acquisition of data, Analysis and interpretation of data, Drafting or revising the article; JGC, Performed experiments, Acquisition of data, Analysis and interpretation of data; DG, TDT, Performed experiments, Acquisition of data, Analysis and interpretation of data, Drafting or revising the article; ELS, Performed experiments and advised study design, Conception and design, Analysis and interpretation of data, Drafting or revising the article; SMH, CKG, Designed the studies, Conception and design, Analysis and interpretation of data, Drafting or revising the article

### Author ORCIDs
Christopher K Glass, http://orcid.org/0000-0003-4344-3592

### Ethics
Animal experimentation: All animal experiments were performed in compliance with the ethical standards set forth by UC San Diego's Institutional Annual Care and Use Committee (IUCAC)to minimize pain and suffering under protocol S01015.

# Additional files

### Major datasets
The following dataset was generated:

| Author(s) | Year | Dataset title | Dataset URL | Database, license, and accessibility information |
|---|---|---|---|---|
| Karmel A Allison Erica L Stone, Jana G Collier, David Gosselin, Ty Dale Troutman, Stephen M Hedrick, Christopher K Glass, | 2015 | Affinity and Dose of TCR Engagement Yield Proportional Enhancer and Gene Activity in CD4 + T Cells | http://www.ncbi.nlm.nih.gov/geo/query/acc.cgi?acc=GSE69545 | Publicly available at NCBI Gene Expression Omnibus (accession no: GSE69545) |

The following previously published datasets were used:

| Author(s) | Year | Dataset title | Dataset URL | Database, license, and accessibility information |
|---|---|---|---|---|
| Chikuma S Suita N, Okazaki IM, Shibayama S, Honjo T, | 2012 | Comparison of gene expression in Wild type and T cell-specific conditional Trim28 KO in TCR stimulated and un-stimulated naive CD4 positive and T regulatory cells | http://www.ncbi.nlm.nih.gov/geo/query/acc.cgi?acc=GSE32224 | Publicly available at NCBI Gene Expression Omnibus (accession no: GSE32224) |
| Wei G Wei L, Zhu J, Zang C, Hu-Li J, Yao Z, Cui K, Kanno Y, Roh TY, Watford WT, Schones DE, Peng W, Sun HW, Paul WE, O'Shea JJ, Zhao K, | 2009 | Epigenetic Mechanisms Underlie T Cell Plasticity | http://www.ncbi.nlm.nih.gov/geo/query/acc.cgi?acc=GSE14308 | Publicly available at NCBI Gene Expression Omnibus (accession no: GSE14308) |

| Author(s) | Year | Title | URL | Availability |
|---|---|---|---|---|
| Wherry EJ Crawford A, Angelosanto J, | 2014 | Longitudinal expression data from CD4+ T cells responding to LCMV-Armstrong or LCMV-Clone 13 | http://www.ncbi.nlm.nih.gov/geo/query/acc.cgi?acc=GSE41866 | Publicly available at NCBI Gene Expression Omnibus (accession no: GSE41866) |
| Wakamatsu E Mathis D, Benoist C, | 2012 | Gene expression profile of conventional T cells (Tconv) and regulatory T cells (Treg) stimulated with anti-costimulatory molecule antibodies | http://www.ncbi.nlm.nih.gov/geo/query/acc.cgi?acc=GSE42276 | Publicly available at NCBI Gene Expression Omnibus (accession no: GSE42276) |
| Wei L Tan C, Gery I, | 2014 | Phenotypes of Th lineages generated by the commonly used activation with anti-CD3/CD28 antibodies differ from those generated by the physiological activation with the specific antigen | http://www.ncbi.nlm.nih.gov/geo/query/acc.cgi?acc=GSE54938 | Publicly available at NCBI Gene Expression Omnibus (accession no: GSE54938) |
| Mostafavi S Ortiz-Lopez A, Mathis D, Benoist C, | 2014 | Gene expression data for 39 inbred mice strains for CD4+ T cells | http://www.ncbi.nlm.nih.gov/geo/query/acc.cgi?acc=GSE60337 | Publicly available at NCBI Gene Expression Omnibus (accession no: GSE60337) |
| Li P Spolski R, Liao W, Wang L, Murphy TL, Murphy KM, Leonard WJ, | 2012 | BATF-JUN is critical for IRF4-mediated transcription in T cells | http://www.ncbi.nlm.nih.gov/geo/query/acc.cgi?acc=GSE39756 | Publicly available at NCBI Gene Expression Omnibus (accession no: GSE39756) |
| Mishra BP Zaffuto KM, Artinger EL, Org T, Mikkola HK, Cheng C, Djabali M, Ernst P, | 2014 | The histone methyltransferase activity of MLL1 is dispensable for hematopoiesis and leukemogenesis | http://www.ncbi.nlm.nih.gov/geo/query/acc.cgi?acc=GSE56456 | Publicly available at NCBI Gene Expression Omnibus (accession no: GSE56456) |
| Zhang JA Mortazavi A, Williams BA, Wold BJ, Rothenberg EV, | 2012 | Dynamic Transformations of Genome-wide Epigenetic Marking and Transcriptional Control Establish T Cell Identity [ChIP-Seq] | http://www.ncbi.nlm.nih.gov/geo/query/acc.cgi?acc=GSE31233 | Publicly available at NCBI Gene Expression Omnibus (accession no: GSE31233) |
| Vahedi G Takahashi H, Nakayamada S, Sun H, Sartorelli V, Kanno Y, O'Shea JJ, | 2012 | STATs Shape the Active Enhancer Landscape of T Cell Populations | http://www.ncbi.nlm.nih.gov/geo/query/acc.cgi?acc=GSE40463 | Publicly available at NCBI Gene Expression Omnibus (accession no: GSE40463) |
| Mikkelsen TS Xu Z, Gimble JM, Lander ES, Rosen ED, | 2010 | Epigenomic profiling of 3T3-L1 adipogenesis | http://www.ncbi.nlm.nih.gov/geo/query/acc.cgi?acc=GSE21365 | Publicly available at NCBI Gene Expression Omnibus (accession no: GSE21365) |
| Buecker C | 2014 | Reorganization of enhancer patterns in transition from naïve to primed pluripotency (ChIP-seq) | http://www.ncbi.nlm.nih.gov/geo/query/acc.cgi?acc=GSE56098 | Publicly available at NCBI Gene Expression Omnibus (accession no: GSE56098) |
| Heinz S Benner C, Spann N, Bertolino E, Lin YC, Laslo P, Cheng JX, Murre C, Singh H, Glass CK, | 2010 | Simple combinations of lineage-determining transcription factors prime cis-regulatory elements required for macrophage and B cell identities | http://www.ncbi.nlm.nih.gov/geo/query/acc.cgi?acc=GSE21512 | Publicly available at NCBI Gene Expression Omnibus (accession no: GSE21512) |

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
