## [Decision Letter]

Thank you for submitting your work entitled "Affinity and Dose of TCR Engagement Yield Proportional Enhancer and Gene Activity in CD4^+^ T Cells" for peer review at *eLife*. Your submission has been favorably evaluated by Tadatsugu Taniguchi (Senior editor), a Reviewing editor, and three reviewers.

The reviewers have discussed the reviews with one another and the Reviewing editor has drafted this decision to help you prepare a revised submission.

Your paper has been seen by a Reviewing Editor and three expert referees. While all believe that the topic is of importance and that the data sets you have generated are of substantial value to the field, there are major concerns about the core conclusions of your study that prevent the paper from being accepted in its present form. Because all the reviews contain important comments and specific requests for new studies, in two cases to clarify the central issue of how T cells respond to various strengths of stimulation (variation per cell in response or variation in the fraction of responding cells) and in another to strengthen the genomic/epigenetic analyses, we have included these reviews verbatim rather than in the usual *eLife* integrated format.

If new experiments and data can be generated to address these issues and a revised paper submitted within the 2 month-window permitted by *eLife* policy, it will be reviewed as a revised submission. If the new work and resubmission takes longer, the paper will be considered a new paper, but we will endeavor to have the same individuals examine the new manuscript to provide consistency in the evaluation process. It may be helpful to address a letter to the Senior Editor addressing what you believe you can achieve in a reasonable length of time to address the criticisms raised by the reviewers.

*Reviewer #1:*

In this manuscript the authors use a well-studied model system in which primary mouse CD4^+^ T cells expressing the transgenic AND TCR are stimulated with antigen presenting cells presenting pigeon cytochrome c (PCC) or variant peptides with a range of affinities for the TCR. While several aspects of T cell activation have been described as digital or analog, the authors contend that this dichotomy is too simple. They show at a single time point that the MFI of CD69 and CD25 of responding cells correlated with the relative strengths of TCR stimulation. To determine how titration of ligand affinity affected the transcriptional responses, the authors used RNA-seq and principal component analysis to identify a subset of "activation signature genes" which responded in a graded manner to peptide affinity and dose. ChIP-seq was used to compare the extent of epigenetic tagging of enhancers and showed that the degree of H3K4me2 tagging of activation signature genes increased in a graded manner in response to changes in TCR stimulation strength. Motif analysis also revealed that enhancers that became more active with TCR signal strength were enriched for AP-1 and NFkB sites. To determine how this subset of activation signature genes responded to perturbation of the Ras/Raf/Mek/Erk pathway, the authors performed ChIP-seq analysis on T cells stimulated with partial Mek inhibition. Consistent with the top 10% of PC1 genes having a bias toward genes containing AP-1 sites, the activation signature genes were appreciably more susceptible to Mek inhibition than the bottom 10%. The authors conclude that the ERK pathway is responsible for translating varying strengths of TCR stimuli into similarly graded enhancer activity.

In this article the authors raise an important question. However, the conclusions drawn in this study, while plausible, lack sufficient supporting experimental evidence.

1) The major design flaw of the RNA-seq and ChIP-seq experiments is that the source RNA or DNA analyzed was pooled from an entire population of stimulated cells (including for example both CD69- and CD69+ cells). As a result, it is not possible to determine whether the graded gene expression/epigenetic changes occur due to graded responses homogenously in every cell, or due to decreasing frequencies of responding cells as the affinity of stimulating peptide is decreased. In Figure 1 the authors demonstrate this heterogeneity at a single cell level that decreasing the potency of TCR stimulus decreases the frequency of cells that express CD69 and CD25. This is not to say that there may be graded responses amongst the responding cells. The data generated by flow cytometry in Figure 1 and Figure 6 demonstrate that the MFI of CD25, CD69 and phospho-ERK can be dose dependent, when gated on the responding cells. One should note, however, that they only do their flow analysis at a single time point and it is not clear that maximal responses have been reached at lower doses of peptides or with weaker peptides. Regardless, however, RNA-seq and ChIP-seq analyses on bulk populations cannot formally discriminate whether the signals detected come from each cell equally or from a subpopulation of cells. A more informative experiment would involve sorting the CD69+ cells after stimulation with varying concentrations/affinities of peptide, and then performing RNA-seq or ChIP-seq.

2) The Abstract and the Discussion imply that ERK activation is the predominant pathway that accounts for inducing graded enhancer activity. However, this ignores other known pathways that are likely to contribute to activation signature gene transcription, such as NFAT and NFkB. Indeed, NFkB sites appeared to be enriched in the activation signature genes as well (Figure 4).

*Reviewer #2:*

Understanding how T lymphocytes respond to antigens of varied doses or quantities remains a critical issue in immunology. Indeed, much biochemical effort has been focused on the first minutes of T cell activation (e.g. biophysics of TCR/pMHC interaction, or dynamics of signaling response or synapse formation). In these settings, T cell activation (as measured by NFAT translocation or ERK phosphorylation among others) was found to be essentially digital, with the frequency of cells getting activated increasing with increasing doses or increasing quality of antigens. On the other hand, the typical degree of activation (amongst activated cells, as measured by the mode of ppERK or NFAT activation) was found to be essentially constant on short timescale (e.g. t<30min), although recent work in the EGFR pathway argued for frequency encoding of activation strength (Albeck et al., 2013). T cell activation and regulation of immune responses occur over longer timescales (> hr): this opens up opportunities for T cells to register, in an analog manner, quantity and quality of antigens, while short-term responses were essentially digital.

In this article, Allison et al. report that early cellular markers of T cell activation (e.g. CD25 and CD69 upregulation) are indeed bimodal yet, the mode of protein abundances amongst activated T cells tracks the strength of antigen stimuli (quality and quantity). Subsequent analysis tracks alterations in the landscape of epigenetic marks to demonstrate their correlation with the strength of antigenic stimulation.

One key issue in this study is that changes in the amplitude of response amongst activated cells (analog mode) occur concomitantly with changes in the frequency of cells getting activated (digital mode), for different antigen strength (cf Figure 1). The authors argue that the dominant mode of regulation is the amplitude of signaling amongst activated cells. This conclusion is supported qualitatively by the application of MEK inhibitors (Figure 6), although the authors fail to report that the impact of MEK inhibition is solely on the mode of ERK phosphorylation and not on the frequency of T cell activation. Yet, most outputs are measured in bulk (e.g. by sequencing of a population of sorted cells) except for *Tbet* and *Irf4* (Figure 2). Thus changes in gene regulation may be dominated by the changes in frequency of activated cells rather than by changes in the modes of activated cells. Overall, a direct quantitative test of this observation is not carried out: changes in frequency of activated T cells and mode of gene upregulation are confounding consequences of antigenic activation that must be better deconvolved at the individual cell level.

The paper also makes a strong case about the ability of T cells to register the strength of antigenic stimuli in terms of gene regulation. Using publicly available data and new datasets acquired in-house, the authors derive a gene signature that encompasses the dominant mode of variation of gene up/down regulation in T lymphocytes. Surprisingly, a single score derived from a simple principal component analysis is shown to encompass almost completely the variability of T cell gene regulation (at homeostasis or under activation). Moreover, this PCA score is shown to report back the strength of activation in different settings (with/without costimulation, with/without involvement of Trim28 etc.): this result is striking as it implies that T cell activation can be quantified as the sum of input signal -a similar result was recently reported by the Hodgkin group (Marchingo et al., 2014). The results and analysis presented here by Allison et al. in terms of gene regulation is very exhaustive and adds to our quantitative understanding of T cell activation.

Allison et al. also report that constitutive levels of activation amongst CD4^+^ T cells (isolated from different mouse strains) vary dramatically and a hierarchy among them can be established based on the universal PCA score derived from T cell activation. This is an interesting observation that would require further investigation: does it imply stronger or weaker responsiveness to antigen stimulation? This issue of potential tuning to constitutive TCR stimulation is long standing (Mandl et al., 2013), and possibly beyond the scope of this study. Still, a more quantitative analysis is warranted to test the strength of this quantitative correlation e.g. using partial least square regression to identify the latent variables (antigen quantity and quality, frequency of response, mode of response) that best account for the measured variability in gene output (Kemp et al., 2007).

Overall, this study reports interesting observations related to quantitative aspects of T cell activation in terms of global gene regulation. Additional effort to rigorously quantify the impact of antigen strength would help deconvolve how frequency and mode of activation impact overall gene regulation at the individual cell level.

References:

Albeck, J.G., Mills, G.B., and Brugge, J.S. (2013). Frequency-modulated pulses of ERK activity transmit quantitative proliferation signals. Molecular cell 49, 249-261.

Kemp, M.L., Wille, L., Lewis, C.L., Nicholson, L.B., and Lauffenburger, D.A. (2007). Quantitative network signal combinations downstream of TCR activation can predict IL-2 production response. Journal of immunology 178, 4984-4992.

Mandl, J.N., Monteiro, J.P., Vrisekoop, N., and Germain, R.N. (2013). T cell-positive selection uses self-ligand binding strength to optimize repertoire recognition of foreign antigens. Immunity 38, 263-274.

Marchingo, J.M., Kan, A., Sutherland, R.M., Duffy, K.R., Wellard, C.J., Belz, G.T., Lew, A.M., Dowling, M.R., Heinzel, S., and Hodgkin, P.D. (2014). T cell signaling. Antigen affinity, costimulation, and cytokine inputs sum linearly to amplify T cell expansion. Science 346, 1123-1127.

*Reviewer #2 (Additional data files and statistical comments):*

Better statistical tests to deconvolve frequency and mode of activation and their impact on gene regulation are warranted.

*Reviewer #3:*

In this study, Allison et al. characterize gene expression and histone modifications in CD4^+^ T-cells subjected to varying levels TCR engagement. The authors use PCA to identify an "activation gene signature" and define an "activation score", concordant with traditional measures of T-cell activation level, and use their enrichment score to re-analyze existing data sets. They also use this approach to analyze the consequence of TCR activation in different mouse strains. They find antigen concentration-dependent variation in histone modification (H3K27ac, H3K4me2) near activation signature genes. Finally, the authors find that MEK inhibition specifically reduces the expression of activation genes. The authors conclude that whereas TCR engagement can be viewed as a digital signal, the ERK pathway translates TCR activation to graded gene activation.

The manuscript reports one of the first genome-wide studies of transcriptional regulation in response to varying levels of stimulation coupled with associated epigenetic changes. As such, this will be of interest to both the T-lymphocyte and broader transcriptional regulation communities, and will also be a useful resource for future investigation. However, there are a number of issues that should be resolved.

For many of the authors’ observations, the authors should present a control where expression levels (or fold-change upon stimulation) are considered. For example, when comparing properties of the top 10% vs bottom 10% of PC1 genes (e.g., histone modification levels, super-enhancer levels, or the effect of MEK inhibition), an alternative to the authors' conclusion that activation signature genes are functionally distinct from non-activation genes is that these features are simply proportional to expression level. Another example of this potential confounding effect is the AP-1 motif frequency in the top-10% vs. bottom-10% of PC1 genes – this might reflect expression level.

To account for differences in expression level, the authors could sample from the set of top and bottom activation genes such that their distributions of expression levels match, and then compare the genomic features between these expression-controlled samples. Some of the reported effects arise from differences in expression level and that some of the authors' observations recapitulate well-established relationships between histone modification levels and expression levels.

In principle, the activation score could be a useful tool; however, what is the evidence that the activation score is a more accurate measure than expression levels of individual well-established activation markers (e.g. CD69, CD25, etc.)? In what way is it superior to these other markers?

Why is a T-cell-specific ontology term not found in the "activation gene signature"? What are the terms enriched in the bottom 10% of activation signature genes? The details of the ontology search should be clarified: did the authors compare annotations of the top-10% to those of the bottom-10% genes? or top-10% genes vs. all genes – in which case there will of course be a bias towards T-cell function because they imposed a minimum expression threshold (10 RPKM) before performing PCA to select signature genes. The authors should also provide sufficient details about the specific random model used whenever they describe a p-value to assess the significance of an observation.

It is well-appreciated in the T cell activation literature that peptide concentration influences Th1/Th2 skewing (e.g. Rogers and Croft, 1999). It is curious that the authors did not choose to examine this outcome. Were "Th2" genes preferentially enriched with low-dose, low-affinity peptides? Likewise, some mouse strains are more Th2-ish than others. Was this evident in the transcriptomes?

The strategy to isolate naïve T cells is problematic. It's not a problem generally since TCR transgenic mice are used. However, it probably is a problem for the experiment shown in Figure 3. This experiment may very well reflect contaminating memory cells. Presumably, these represent cells directly analyzed following isolation. The authors should show their isolation technique results in a truly naïve population of cells.

It is not clear from the Methods how many replicates were obtained in RNA-seq and ChIP-seq experiments. In general, the figure legends and Methods should more clearly state the replicates and independent experiments performed.

The histone ChIP-seq experiments refer to "across the five conditions". Presumably this refers to unstimulated cells and the two concentrations of the two peptides. However, the preceding figure includes lots of conditions; this will be confusing for readers. There is no presentation of global data depicting responses to both peptides and doses. More information on numbers of peaks, quality control and overlap (Venn diagrams etc.) would be of interest.

Readers may find Figure 4 confusing. The text refers to AP1 motifs and the figure depicts BATF. BATF, of course, is a member of the AP1 family; however, not all readers may immediately understand this. The figure legend is clear, but the text may still be confusing. The text also tacitly equates AP1 and *Batf*, but this is obviously not the case. Fos/Jun are players in TCR signaling. It is understandable that the authors have used existing *Batf* ChIP-seq data, but other AP1 family members undoubtedly contribute to activation.

Figure 4 is also confusing. Readers may not understand right away that the legend indicates solid and dashed lines. It is an important piece of data, so the authors should make it easier for readers to get this right away. The choice of colors is also confusing (purple refers to *Batf* and CTCF). This figure is more complicated than it needs to be. Is 0.1 μm PCC used? Is 1 μm PCC not used in Figure A? The authors should check all the figures for labeling errors, e.g.: Figure 1: green 100 μm curve is labeled PCC instead of K99A. Figure 1: x-axis labels are missing decimal points. The choice of CTCF is also of interest, insofar as one would not necessarily expect that it would be a TF that would be responsive to graded signals.

Other points:

In Figure 2 it would be desirable to show both MFI and% positive cells. Given the effect of peptide concentration on Th1/Th2 skewing, showing GATA3 as well might be illuminating.

Figure 5 – more explanation of Pleckstrin homology genes and the potential relevance to T cell activation is warranted.

[Editors' note: further revisions were requested prior to acceptance, as described below.]

Thank you for resubmitting your work entitled "Affinity and Dose of TCR Engagement Yield Proportional Enhancer and Gene Activity in CD4^+^ T Cells" for further consideration at *eLife*. Your revised article has been favorably evaluated by Tadatsugu Taniguchi (Senior editor), a Reviewing editor, and three reviewers.

The manuscript has been improved but there are some remaining issues that need to be addressed before acceptance, as outlined below:

Two of the reviewers felt that some primary data were needed to help the readers evaluate important new studies now included in the revised manuscript. There were also some important discussion issues that should be addressed.

*Reviewer #1:*

In the original manuscript by Allison et al., a major experimental design flaw pointed out by two reviewers was that the source material for the RNA-Seq and ChIP-seq data studies was from a mixed population of responding (CD69+) and non-responding (CD69-) T cells, rather than a purified population of responding (CD69+) cells. In the revised manuscript, the authors have added new experimental data that address this major concern. Figure 3 shows graded expression of a single example, *Irf4* transcripts, amongst the CD69+ cells that respond to APC/peptides of varying affinity, and the rest of Figure 3 shows RNA-Seq data from the CD69+ cells sorted from a bulk population of activated T cells. The results of both approaches are consistent with the authors' conclusions that there can be graded transcriptional signatures on a per-cell basis. Whereas the RNA-Seq data largely satisfy this concern, there is still some uncertainty about the results presented by the RNA-flow in Figure 3. The authors should include data in the figure for some "loading control" type controls, or alternatively, other examples. Specifically, the concern is whether most or all transcripts exhibit a similar pattern (graded, and increases with increased peptide affinity) using this technique, which is relatively new and not yet widely used in the field.

Two other concerns about the manuscript that could be addressed in the Discussion:

1) The authors should discuss the work in Huang et al., 2013 a bit more in which the Davis lab showed that single T cell engagement of different numbers of identical peptide agonists (titration of signal strength) resulted in digital responses read out as increasing numbers of T cells producing the same amount of cytokine. Different levels of responses were seen when naïve cells were compared with blast cell responses – at the single cell level.

Second, the authors seem not to have considered different temporal responses. Analysis at a single time point eliminates the potential for the cumulative effects of weak signaling, particularly if there is asynchrony in the population. Strong signals tend to be more synchronous.

*Reviewer #2:*

This resubmission includes an additional set of experiments that goes a long way towards addressing the previous reviews. The authors use RNAseq on activated/sorted T cells to demonstrate that gene regulation does scale with TCR activation, even within cells that are activated.

The authors attempted to resolve the issue of digitalness/analogness of the response by performing RNA flow cytometry (e.g. measuring IRF4). It reads like the dynamics range and signal-to-noise ration of detecting *irf4* mRNA was insufficient to rule out bimodality. Still the trick of sorting and analyzing single cells does go a long way towards establishing the ability of activated T cells to scale their response to the strength of antigen activation.

I would recommend that the histograms or cumulative distribution function for *irf4* mRNA should be presented to let readers assess the lack of bimodality. I am surprised that the 10 fold increase (between activation with no peptide activation or with PCC at 10µM – Figure 3) is not sufficient for resolution.

I appreciate the care and additional work carried out since the last submission.

Reviewer #3:

The revised manuscript satisfactorily addresses my concerns.

---

## [Author Response]

Reviewer #1:

In this article the authors raise an important question. However, the conclusions drawn in this study, while plausible, lack sufficient supporting experimental evidence.

1) The major design flaw of the RNA-seq and ChIP-seq experiments is that the source RNA or DNA analyzed was pooled from an entire population of stimulated cells (including for example both CD69- and CD69+ cells). As a result, it is not possible to determine whether the graded gene expression/epigenetic changes occur due to graded responses homogenously in every cell, or due to decreasing frequencies of responding cells as the affinity of stimulating peptide is decreased. In Figure 1 the authors demonstrate this heterogeneity at a single cell level that decreasing the potency of TCR stimulus decreases the frequency of cells that express CD69 and CD25. This is not to say that there may be graded responses amongst the responding cells. The data generated by flow cytometry in Figure 1 and Figure 6 demonstrate that the MFI of CD25, CD69 and phospho-ERK can be dose dependent, when gated on the responding cells. One should note, however, that they only do their flow analysis at a single time point and it is not clear that maximal responses have been reached at lower doses of peptides or with weaker peptides. Regardless, however, RNA-seq and ChIP-seq analyses on bulk populations cannot formally discriminate whether the signals detected come from each cell equally or from a subpopulation of cells. A more informative experiment would involve sorting the CD69+ cells after stimulation with varying concentrations/affinities of peptide, and then performing RNA-seq or ChIP-seq.

We agree with Reviewer #1 that while flow cytometry experiments indicated graded responses of protein expression for selected markers in responding cells, this pattern is not established for mRNA by the RNA-Seq studies of bulk populations of cells. The issue of bulk vs activated cells is also raised by Reviewer #2. To address this concern, we first performed mRNA flow cytometry, in which fluorescent probes are hybridized to target mRNA and analyzed with traditional flow cytometry, resulting in single cell mRNA data. RNA flow experiments demonstrated a graded response in IRF4 mRNA in the responding cells. These results are presented in new Figure 3 and Figure 3—figure supplement 1. A limitation of the RNA flow technology in our hands was that the dynamic range of IRF4 induction was much lower than that observed using RNA-Sequencing, likely due to the technology used to amplify and detect the mRNA target. Therefore, to confirm this observation and look genome-wide, we conducted RNA-sequencing on CD4^+^ T cells treated as before, but sorted by expression of the activation marker CD69, thereby selectively analyzing responder cells. Although there were interesting differences from RNA-Seq results obtained in the bulk population as elaborated on below, we found that the first principal component again reflected the strength of signaling, with 10μM K99A the leftmost sample along PC1, and 1μM PCC the rightmost sample. This result is shown in new Figure 3. This is further illustrated by the mean expression levels of mRNAs corresponding to the upper 10% of PC1, which exhibit a clear dependence on dose and affinity (Figure 3). The dominant functional annotations are associated with protein synthesis and ribosome biogenesis (3E), also consistent with the functional annotations associated with the corresponding genes identified in the bulk RNA-seq experiments (Figure 2). Examples of genes involved in translational initiation or ribosome biogenesis that exhibit analogue responses are illustrated in Figure 2 and G. We also observed a cluster of genes that exhibited a digital response – i.e., maximal or near maximal expression in the small number of CD69 responding cells recovered after treatment with the low dose of the low affinity peptide (10μM K99A) in comparison to expression in the much larger number of cells recovered after treatment with the high dose of the high affinity ligand (1μM PCC). The top 100 genes from this cluster are illustrated in Figure 3. Interestingly, this cluster of genes is enriched for functional annotations for immune response (Figure 3), with representative genes illustrated in Figure 3. Therefore, analogue signals for these mRNAs in the bulk RNA sequencing experiment primarily reflect differences in frequencies of responding cells. This implies first that the graded levels of immune response genes seen at the protein level in single cells is a result of post-transcriptional regulation, rather than transcriptional regulation. Mechanisms that account for both digital and analogue responses within responsive cells remain to be established, but may be partly due to differences in kinetics, which are beyond the scope of the present studies. Nevertheless, we think that these results substantially strengthen and extend our original conclusions that both the frequency of responding cells and the per- cell strength of response contribute to the population response to antigens of varying concentration and/or affinity.

2) The Abstract and the Discussion imply that ERK activation is the predominant pathway that accounts for inducing graded enhancer activity. However, this ignores other known pathways that are likely to contribute to activation signature gene transcription, such as NFAT and NFkB. Indeed, NFkB sites appeared to be enriched in the activation signature genes as well (Figure 4).

We agree with this point. These TFs clearly play crucial roles in activation of T cells, and we cannot, based on the current study, rank the importance of these TFs. Instead, we would advocate for subsequent studies to investigate the role of these TFs in fine-tuning T cell signaling. We therefore revised the last sentence of the Abstract to:

“Finally, we show that graded expression of activation genes depends on ERK pathway activation, suggesting that an ERK-AP-1 axis plays an important role in translating TCR signal strength into proportional activation of enhancers and genes essential for T cell function.”

In addition, we added the following clarification to the Discussion section:

“Notably, NF-κB is one of many transcription factors known to play important roles in T cell biology, and indeed we find an NF-κB motif enriched among enhancers that are responsive to stimulation (Figure 4). Further research as to the relationship between the strength of TCR signaling and other signal-dependent transcription factors such as NF-κB and NFAT is warranted.”

Reviewer #2:

One key issue in this study is that changes in the amplitude of response amongst activated cells (analog mode) occur concomitantly with changes in the frequency of cells getting activated (digital mode), for different antigen strength (cf Figure 1). The authors argue that the dominant mode of regulation is the amplitude of signaling amongst activated cells. This conclusion is supported qualitatively by the application of MEK inhibitors (Figure 6), although the authors fail to report that the impact of MEK inhibition is solely on the mode of ERK phosphorylation and not on the frequency of T cell activation. Yet, most outputs are measured in bulk (e.g. by sequencing of a population of sorted cells) except for Tbet and Irf4 (Figure 2). Thus changes in gene regulation may be dominated by the changes in frequency of activated cells rather than by changes in the modes of activated cells. Overall, a direct quantitative test of this observation is not carried out: changes in frequency of activated T cells and mode of gene upregulation are confounding consequences of antigenic activation that must be better deconvolved at the individual cell level.

We agree with the major concerns of Reviewer #2, some of which are also raised by Reviewer #1. We would not rank amplitude over frequency, but instead wish to show that amplitude contributes to the strength of signaling in CD4^+^ T cells. Indeed, in the case of MEK inhibition, frequency is affected. We provide a new panel in revised Figure 7—figure supplement 1 to make this point. However, the single-cell expression level changes of activation markers in response to MEK inhibition (Figure 7—figure supplement 1) indicate that both frequency and per-cell differences play a role in the total population output. We also reference Albeck et al. in the revised manuscript.

We also agree that the manuscript would be strengthened by single cell data corresponding to the bulk mRNA data. To address this concern, we performed mRNA flow cytometry. As detailed in the response to Reviewer #1, these experiments demonstrated a graded response in IRF4 mRNA in CD69-positive cells. The data is included in new Figure 3. We also considered performing RNA sequencing in individual responding cells. However, this methodology as currently employed is unable to detect most transcripts in a particular cell, let alone provide quantitative measures of transcript abundance. As an alternative, we performed RNA-Seq of responding cells for low and high doses of K99A and PCC. These studies, which are detailed in the response to Reviewer #1, confirmed a strong analogue component of mRNA expression in responding cells (new Figure 3). Genes exhibiting an analogue component are enriched for functional annotations related to protein translation, consistent with our original findings (new Figure 3). Interestingly, we also identified a cluster of genes exhibiting a digital response in responding cells. These genes are enriched for functional annotations related to immune response (new Figure 3). Therefore, analogue signals for these mRNAs in the bulk RNA sequencing experiment primarily reflect differences in frequencies of responding cells. This implies first that the graded levels of immune response genes seen at the protein level in single cells is a result of post- transcriptional regulation, rather than transcriptional regulation. We think that these results substantially strengthen and extend our original conclusions that both the frequency of responding cells and the per- cell strength of response contribute to the population response to antigens of varying concentration and/or affinity and we thank Reviewer #2 for prompting these experiments.

The paper also makes a strong case about the ability of T cells to register the strength of antigenic stimuli in terms of gene regulation. Using publicly available data and new datasets acquired in-house, the authors derive a gene signature that encompasses the dominant mode of variation of gene up/down regulation in T lymphocytes. Surprisingly, a single score derived from a simple principal component analysis is shown to encompass almost completely the variability of T cell gene regulation (at homeostasis or under activation). Moreover, this PCA score is shown to report back the strength of activation in different settings (with/without costimulation, with/without involvement of Trim28 etc.): this result is striking as it implies that T cell activation can be quantified as the sum of input signal -a similar result was recently reported by the Hodgkin group (Marchingo et al., 2014). The results and analysis presented here by Allison et al. in terms of gene regulation is very exhaustive and adds to our quantitative understanding of T cell activation.

We thank Reviewer #2 for these comments. We include reference to Marchingo et al. in the revised manuscript.

Allison et al. also report that constitutive levels of activation amongst CD4^+^ T cells (isolated from different mouse strains) vary dramatically and a hierarchy among them can be established based on the universal PCA score derived from T cell activation. This is an interesting observation that would require further investigation: does it imply stronger or weaker responsiveness to antigen stimulation? This issue of potential tuning to constitutive TCR stimulation is long standing (Mandl et al., 2013), and possibly beyond the scope of this study. Still, a more quantitative analysis is warranted to test the strength of this quantitative correlation e.g. using partial least square regression to identify the latent variables (antigen quantity and quality, frequency of response, mode of response) that best account for the measured variability in gene output (Kemp et al., 2007).

We thank Reviewer #2 for these comments. With regard to a least squares regression analysis, the inbred strain data was derived from publicly available ImmGen data, which unfortunately does not include stimulated cells. Using our five conditions would give us too few conditions for the number of variables being tested. We agree that while acquiring stimulation data across multiple doses or peptides for the 39 inbred strains is outside the scope of this study, our findings do suggest an interesting approach to using strains to tease out the relative contributions of dose and affinity to strength of activation.

References:

Albeck, J.G., Mills, G.B., and Brugge, J.S. (2013). Frequency-modulated pulses of ERK activity transmit quantitative proliferation signals. Molecular cell 49, 249-261.

Kemp, M.L., Wille, L., Lewis, C.L., Nicholson, L.B., and Lauffenburger, D.A. (2007). Quantitative network signal combinations downstream of TCR activation can predict IL-2 production response. Journal of immunology 178, 4984-4992.

Mandl, J.N., Monteiro, J.P., Vrisekoop, N., and Germain, R.N. (2013). T cell-positive selection uses self-ligand binding strength to optimize repertoire recognition of foreign antigens. Immunity 38, 263-274.

*Marchingo, J.M., Kan, A., Sutherland, R.M., Duffy, K.R., Wellard, C.J., Belz, G.T., Lew, A.M., Dowling, M.R., Heinzel, S., and Hodgkin, P.D. (2014). T cell signaling. Antigen affinity, costimulation, and cytokine inputs sum linearly to amplify T cell expansion. Science 346, 1123-1127.*

Reviewer #2 (Additional data files and statistical comments):

Better statistical tests to deconvolve frequency and mode of activation and their impact on gene regulation are warranted.

In order to better separate the role of frequency from that of amplitude of signal, we conducted single-cell mRNA flow cytometry and sorted RNA-seq, as described above. In addition, to further deconvolute the contributors to strength of activation, we segmented the genes in PC1 according to expression level as suggested by Reviewer #3. These analyses demonstrate that the response to the MEK inhibitor as well as AP-1 binding frequency are not dependent on the expression level of the genes in question (See responses to Reviewer #3 below).

Reviewer #3:

For many of the authors’ observations, the authors should present a control where expression levels (or fold-change upon stimulation) are considered. For example, when comparing properties of the top 10% vs bottom 10% of PC1 genes (e.g., histone modification levels, super-enhancer levels, or the effect of MEK inhibition), an alternative to the authors' conclusion that activation signature genes are functionally distinct from non-activation genes is that these features are simply proportional to expression level. Another example of this potential confounding effect is the AP-1 motif frequency in the top-10% vs. bottom-10% of PC1 genes – this might reflect expression level.

To account for differences in expression level, the authors could sample from the set of top and bottom activation genes such that their distributions of expression levels match, and then compare the genomic features between these expression-controlled samples. Some of the reported effects arise from differences in expression level and that some of the authors' observations recapitulate well-established relationships between histone modification levels and expression levels.

To investigate the effect of expression level, we took subsets of the top and bottom genes according to RPKM ranges. This greatly reduced the number of genes and related entities that could be compared in any given set, reducing the statistical significance of any conclusions. Nonetheless, RPKM did not have a qualitative effect on the associations found. As can be seen in new Figure 7—figure supplement 2 and Figure 7—figure supplement 3, AP-1 correlates with the PC1 segment across RPKM buckets, and the effect of MEKi on PC1 segments is likewise consistent across buckets.

In principle, the activation score could be a useful tool; however, what is the evidence that the activation score is a more accurate measure than expression levels of individual well-established activation markers (e.g. CD69, CD25, etc.)? In what way is it superior to these other markers?

The most important aspect of the activation score is that it is multidimensional and incorporates information on a broad spectrum of genes that can become activated in response to T cell receptor ligation. In contrast, while specific activation markers provide information on functionally important molecules, they are typically considered to be ‘positive’ or ‘negative’ and do not necessarily reflect the global state of activation. For example, CD69-positive cells responding to the low affinity peptide have a lower activation score than CD69-positive cells responding to the high affinity peptide. We found this to be the case regardless of whether we used bulk population RNA-Seq data or RNA-Seq data obtained from responding cells. This is mostly due to lower levels of expression of genes required for protein translation and ribosomal biogenesis, which are presumably required for proliferation and various T cell effector functions. Thus the activation score provides additional information that is not obtained through measurement of well-established activation markers.

Why is a T-cell-specific ontology term not found in the "activation gene signature"? What are the terms enriched in the bottom 10% of activation signature genes? The details of the ontology search should be clarified: did the authors compare annotations of the top-10% to those of the bottom-10% genes? or top-10% genes vs. all genes – in which case there will of course be a bias towards T-cell function because they imposed a minimum expression threshold (10 RPKM) before performing PCA to select signature genes. The authors should also provide sufficient details about the specific random model used whenever they describe a p-value to assess the significance of an observation.

As to T cell specific terms, some did indeed appear in the GO list, but ontology analysis is a numbers game; there are fewer T cell specific genes among the PC1 set than metabolic and biosynthetic process genes, resulting in immune terms being pushed farther down the list. Several lymphocyte terms- specific terms achieved significant Benjamini-adjusted P-values but were not included in the original table. One problem we became aware of during preparation of the revised version of the manuscript is that the DAVID tool for gene ontology has not been updated in 6 years. We therefore repeated the GO analysis using Metascape, which is updated on a continuous basis. The results are qualitatively similar, but this analysis captures more annotations. Terms related to protein synthesis still dominate the top of the list, but many terms related to immune response are also captured. These new results are provided in the revised Figure 2. In addition, we performed RNA-Seq of responding (CD69+) cells to confirm results of bulk RNA Seq experiments as requested by Reviewers #1 and #2. This experiment also identified biosynthetic process genes as being the most significant in PC1, which demonstrated an analogue response (new Figure 3). However, we also identified a ‘digital’ cluster in the responding population of cells that was not apparent in the bulk sequencing experiment. Interestingly, this cluster of genes is much more enriched for immune response genes, illustrated in new Figure 3. As now described in methods, we ran GO analysis on Metascape using default parameters for transcriptome background and the q statistic for adjusting p-values for multiple testing.

It is well-appreciated in the T cell activation literature that peptide concentration influences Th1/Th2 skewing (e.g. Rogers and Croft, 1999). It is curious that the authors did not choose to examine this outcome. Were "Th2" genes preferentially enriched with low-dose, low-affinity peptides? Likewise, some mouse strains are more Th2-ish than others. Was this evident in the transcriptomes?

We share the reviewer’s interest in the Th1/Th2 dichotomy, and indeed we looked at Gata3 across the conditions via flow cytometry, and we compared expression of many different markers of Th2 cells in the RNA-seq data. We did not observe any significant expression of Gata3 in any of the conditions via intra-cellular flow cytometry. Similarly, all of the Th2 markers we looked at in the high- throughput data showed low RPKM values as well as expression changes that were not significant or were inconsistent across replicates. We posit that this difference versus other references may be the result of the particular low-affinity peptide chosen, as low-affinity responses seem to vary widely in the end phenotype. We note in the revised Discussion that we observe no evidence of Th1/Th2 skewing.

The strategy to isolate naïve T cells is problematic. It's not a problem generally since TCR transgenic mice are used. However, it probably is a problem for the experiment shown in Figure 3. This experiment may very well reflect contaminating memory cells. Presumably, these represent cells directly analyzed following isolation. The authors should show their isolation technique results in a truly naïve population of cells.

Figure 3 is derived from public data published by the ImmGen Consortium. According to the data submission (http://www.ncbi.nlm.nih.gov/geo/query/acc.cgi?acc=GSE60337), the cells were sorted to be CD3+CD4^+^CD62L+, implying that memory cell contamination was likely minimal.

*It is not clear from the methods how many replicates were obtained in RNA-seq and ChIP-seq experiments. In general, the figure legends and methods should more clearly state the replicates and independent experiments performed.*

This was an oversight. We have now included reference to replicate number in the appropriate methods sections of the manuscript, as included below:

“Two replicates of the H3K4me2 ChIP-seq across all five conditions were obtained; three replicates across three conditions and one across all five conditions were obtained for the H3K27Ac ChIP-seq; and two replicates across all five conditions of the RNA-seq were obtained.”

Confirmation of bulk RNA-Seq results using sorted cells was performed for no peptide CD69 negative cells and CD69+ cells for low and high doses of K99A and PCC.

All flow cytometry results shown are representative of at least two biological replicates, and the results shown in Figure 1 were reflected with samples held out of each high-throughput sequencing assay.

The histone ChIP-seq experiments refer to "across the five conditions". Presumably this refers to unstimulated cells and the two concentrations of the two peptides. However, the preceding figure includes lots of conditions; this will be confusing for readers. There is no presentation of global data depicting responses to both peptides and doses. More information on numbers of peaks, quality control and overlap (Venn diagrams etc.) would be of interest.

Venn Diagrams across all five of the conditions prove visually hard to interpret. We instead provide below a table of the pairwise overlap between the peaks called with histone mark. In each cell is the count of peaks where each condition shown has at least 40 tags (normalized). In the diagonal is the total number of peaks with at least 40 tags for the given condition. We include these tables as Table 1 and Table 2. As discussed in the manuscript, the vast majority of peaks are shared, especially for H3K4me2.

*Readers may find Figure 4 confusing. The text refers to AP1 motifs and the figure depicts BATF. BATF, of course, is a member of the AP1 family; however, not all readers may immediately understand this. The figure legend is clear, but the text may still be confusing. The text also tacitly equates AP1 and Batf, but this is obviously not the case. Fos/Jun are players in TCR signaling. It is understandable that the authors have used existing Batf ChIP-seq data, but other AP1 family members undoubtedly contribute to activation.*

Good point – it is easy to forget that not everyone spends all day looking at motif sequence logos. In this case, the motifs are derived from the H3K27Ac chip, and are not specific to any single AP-1 family member. The motif finding software applied the BATF label to this particular motif because its original source for that motif happened to be a BATF chip. However, this motif is identical to the generic AP-1 motif. We therefore changed the name of this motif in Figure 4 (now Figure 5) to an AP-1 motif and corresponding text for ease of interpretation.

Figure 4 is also confusing. Readers may not understand right away that the legend indicates solid and dashed lines. It is an important piece of data, so the authors should make it easier for readers to get this right away. The choice of colors is also confusing (purple refers to Batf and CTCF). This figure is more complicated than it needs to be. Is 0.1 μm PCC used? Is 1 μm PCC not used in Figure A? The authors should check all the figures for labeling errors, e.g.: Figure 1: green 100 μm curve is labeled PCC instead of K99A. Figure 1: x-axis labels are missing decimal points. The choice of CTCF is also of interest, insofar as one would not necessarily expect that it would be a TF that would be responsive to graded signals.

We clarified the legend in Figure 5, Figure 5—figure supplement 1 and 1I by making the dashing more clear. We have also done another pass through the figure labels and corrected the errors noted by the Reviewer. As to the choice of CTCF, the goal was to choose a TF that would not be responsive, as signal at any responsive TF is likely to be heavily confounded by the fact that enhancers are highly interconnected and multiply bound. CTCF is thus a more reliable negative control.

We clarified this in the figure description for these figures:

H. Genome-wide, deposition of H3K27Ac, a marker of transcription factor activity, reflects increasing TCR signal strength at the binding sites of AP-1 family members, including *Batf*. Deposition of H3K27Ac at CTCF, a transcription factor that is not expected to be signal responsive, is shown for comparison in the dashed lines.

Other points:

In Figure 2 it would be desirable to show both MFI and% positive cells. Given the effect of peptide concentration on Th1/Th2 skewing, showing GATA3 as well might be illuminating.

For IRF4 and Tbet (Figure 2), we do not see the bimodal distribution that extracellular markers like CD69 and CD25 show. Thus, the MFI shown represents the entire population. We clarified this in the figure legend:

“As measured by flow cytometry, the geometric MFI of *Tbet* in CD4^+^ cells increases on a per-cell basis with increasing signal strength. Note that MFI of the entire population of CD4^+^ cells is shown, as *Tbet* distribution is unimodal.”

We did not observe consistent GATA3 expression as noted above.

Figure 5 – more explanation of Pleckstrin homology genes and the potential relevance to T cell activation is warranted.

Point taken. We revised the manuscript as follows:

“GO analysis of genes nearby the shared super-enhancers showed enrichment for leukocyte activation genes as well as Pleckstrin homology genes, which are critical components of a number of kinase signaling pathways downstream of the TCR (Figure 5), indicating that super- enhancers in CD4^+^ T cells prime genes important for inflammatory signaling.”

[Editors' note: further revisions were requested prior to acceptance, as described below.]

The manuscript has been improved but there are some remaining issues that need to be addressed before acceptance, as outlined below:

Two of the reviewers felt that some primary data were needed to help the readers evaluate important new studies now included in the revised manuscript. There were also some important discussion issues that should be addressed.

Reviewer #1:

In the original manuscript by Allison et al., a major experimental design flaw pointed out by two reviewers was that the source material for the RNA-Seq and ChIP-seq data studies was from a mixed population of responding (CD69+) and non-responding (CD69-) T cells, rather than a purified population of responding (CD69+) cells. In the revised manuscript, the authors have added new experimental data that address this major concern. Figure 3 shows graded expression of a single example, Irf4 transcripts, amongst the CD69+ cells that respond to APC/peptides of varying affinity, and the rest of Figure 3 shows RNA-Seq data from the CD69+ cells sorted from a bulk population of activated T cells. The results of both approaches are consistent with the authors' conclusions that there can be graded transcriptional signatures on a per-cell basis. Whereas the RNA-Seq data largely satisfy this concern, there is still some uncertainty about the results presented by the RNA-flow in Figure 3. The authors should include data in the figure for some "loading control" type controls, or alternatively, other examples. Specifically, the concern is whether most or all transcripts exhibit a similar pattern (graded, and increases with increased peptide affinity) using this technique, which is relatively new and not yet widely used in the field.

To address this concern and a corresponding concern of Reviewer 2 we now provide RNA-flow data for β actin as a loading control. We added panels to Figure 3—figure supplement 1 illustrating Cumulative Distribution Plots for *Irf4* and β actin (panels B and C) and provide a Mean Fluorescence Intensity plot for β actin (panel D), demonstrating that, in contrast to *Irf4*, β actin mRNA expression is constant across treatment conditions when measured by this assay. These results are also consistent with the independent RNA-Seq experiments of CD69+ cells.

Two other concerns about the manuscript that could be addressed in the Discussion:

1) The authors should discuss the work in Huang et al., 2013 a bit more in which the Davis lab showed that single T cell engagement of different numbers of identical peptide agonists (titration of signal strength) resulted in digital responses read out as increasing numbers of T cells producing the same amount of cytokine. Different levels of responses were seen when naïve cells were compared with blast cell responses – at the single cell level.

We thank reviewer 1 for requesting that we take another look at Huang et al. with respect to both the digital response point and the increased rates of protein production in blasts and memory cells in comparison to naïve cells. This difference could very well be explained by the induction of the protein biosynthetic machinery observed in our studies and is now pointed out in the discussion. In addition, we also note that the RNA-seq analysis of CD69+ cells identified a digital cluster of genes that includes *Tnf* and *Il2*. Bar graphs for these and additional representative mRNAs are now included in Figure 3—figure supplement 1, panel E. Thus, our findings are consistent with those of Huang et al. with respect to both changes in synthetic capacity and digital induction of *Tnf* and *Il2*. It will be of interest to further investigate mechanisms enabling gene-specific analogue or digital responses within the same cell.

Second, the authors seem not to have considered different temporal responses. Analysis at a single time point eliminates the potential for the cumulative effects of weak signaling, particularly if there is asynchrony in the population. Strong signals tend to be more synchronous.

We agree that the single time point eliminates the potential for registering cumulative effects of weak signaling and have added this point to the Discussion.

Reviewer #2:

This resubmission includes an additional set of experiments that goes a long way towards addressing the previous reviews. The authors use RNAseq on activated/sorted T cells to demonstrate that gene regulation does scale with TCR activation, even within cells that are activated.

The authors attempted to resolve the issue of digitalness/analogness of the response by performing RNA flow cytometry (e.g. measuring IRF4). It reads like the dynamics range and signal-to-noise ration of detecting irf4 mRNA was insufficient to rule out bimodality. Still the trick of sorting and analyzing single cells does go a long way towards establishing the ability of activated T cells to scale their response to the strength of antigen activation.

I would recommend that the histograms or cumulative distribution function for irf4 mRNA should be presented to let readers assess the lack of bimodality. I am surprised that the 10 fold increase (between activation with no peptide activation or with PCC at 10µM – Figure 3) is not sufficient for resolution.

To address this concern and a corresponding concern of Reviewer 1 we now provide Cumulative Distribution Plots for *Irf4* and β actin (Figure 3—figure supplement 1, panels B and C) and provide a Mean Fluorescence Intensity plot for β actin (panel D), demonstrating that, in contrast to *Irf4*, β actin mRNA expression is constant across treatment conditions when measured by this assay. These results are also consistent with the independent RNA-Seq experiments of CD69+ cells.